

# Sources and physicochemical characteristics of black carbon aerosol in the southeastern Tibetan Plateau: internal mixing enhances light absorption

Qiyuan Wang[1*], Junji Cao[1,2*], Yongming Han[1,3], Jie Tian[4], Chongshu Zhu[1], Yonggang Zhang[1], Ningning
Zhang[1], Zhenxing Shen[4], Haiyan Ni[1], Shuyu Zhao[1], Jiarui Wu[1]

[1]Key Laboratory of Aerosol Chemistry and Physics, State Key Laboratory of Loess and Quaternary Geology, Institute of Earth
Environment, Chinese Academy of Sciences, Xi'an, 710061, China
[2]Institute of Global Environmental Change, Xi'an Jiaotong University, Xi'an, 710049, China
[3]School of Human Settlements and Civil Engineering, Xi'an Jiaotong University, Xi'an, 710049, China
[4]Department of Environmental Sciences and Engineering, Xi'an Jiaotong University, Xi'an, 710049, China

*Correspondence to*: Junji Cao (cao@loess.llqg.ac.cn) and Qiyuan Wang (wangqy@ieecas.cn)

**Abstract.** Black carbon (BC) aerosol over the Tibetan Plateau (TP) has important effects on regional climate and hydrological processes. An intensive measurement campaign was conducted at Lulang (~3300 m above sea level), southeastern TP, from September to October 2015 to investigate the sources and physicochemical characteristics of refractory BC (rBC) aerosol. The
grand average rBC mass concentration was $0.31 \pm 0.55 \ \mu g \ m^{-3}$, which is higher than most BC results for the TP. A clear diurnal cycle in rBC showed high values in the morning and low values in the afternoon. A bivariate polar plot showed that rBC loadings were affected by wind speed and direction, and it was used to infer the dominant transport directions. The estimated net surface transport intensity for rBC was $+0.05 \pm 0.29 \ \mu g \ s^{-1} \ m^{-2}$, indicating that stronger transport from outside the TP compared with the interior of the TP. Cluster analysis and a concentration-weighted trajectory model indicated that emissions
from north India had important contribution to the high rBC, but the effects of internal TP sources cannot be overlooked. The average mass median diameter (MMD) of rBC was $160 \pm 23$ nm, and it was smaller size on rainy days (145 nm) compared with non-rainy days (164 nm). The average number fraction of thickly-coated rBC ($F_{rBC}$) was $39 \pm 8\%$, and it increased with the enhanced $O_3$ mixing ratio from 10:00–14:00 local time, indicating that the photochemical oxidation played an important role in forming rBC coatings. The slope of a linear regression of $F_{rBC}$ versus $O_3$ was 0.44 % ppb$^{-1}$, which was higher than the
value of 0.24 % ppb$^{-1}$ observed at Qinghai Lake, northeastern TP, suggesting that the rBC particles at Lulang were more rapidly affected by internal mixing than those at Qinghai Lake. The average rBC absorption enhancement ($E_{abs}$) was estimated to be 1.8, suggesting that light absorption by coated rBC particles was greater than that for the uncoated ones. The $E_{abs}$ was strongly positively correlated with the $F_{rBC}$, indicating an amplification of light absorption for internally-mixed rBC. When rBC core <170 nm, the $E_{abs}$ was negatively correlated with MMD, but it was nearly constant for rBC core >170 nm. Our study provides
insight into the sources and evolution of rBC aerosol in the TP, and is useful for further modelling studies improving precise of evaluating the radiative forcing of carbonaceous aerosols in this area.



# 1 Introduction

The Tibetan Plateau (TP) is the world's largest high-elevation region, and it holds the largest ice mass on the planet outside the polar regions and is sometimes called the Earth's "Third Pole" (Yao et al., 2008). The snow and associated glacial meltwater on the TP provides fresh water for drinking and irrigation for more than 1 billion people in downstream regions (Immerzeel et al., 2010). The TP exerts significant thermal and dynamic impacts on hydrological processes in South and East Asia. For example, changes in the area covered by glaciers and snowpacks on the TP affect the heat fluxes and water exchange between the atmosphere and the earth's surface, and that, in turn, affects the atmospheric circulation associated with the Asian Monsoon System (Lau and Kim, 2006). Glaciers respond sensitively to climate change (Dyurgerov and Meier, 2000), and recent observations have shown a continuing retreat in Tibetan glaciers (e.g., Xu et al., 2009; Yao et al., 2012; Zhang et al., 2012; Loibl et al., 2014; Kang et al., 2015; Huintjes et al., 2016; Li et al., 2016; Ke et al., 2017). For instance, Yao et al. (2012) reviewed the status of glaciers on the TP and surrounding areas over the past 30 years, and these authors reported systematic differences from region to region. The greatest reduction in glacial length and area and the most negative mass balance occurred in the Himalayas (excluding the Karakorum).

The past few decades have witnessed rapid growth in the human population, industrialization, and urbanization over South and East Asia, and this growth has led to widespread air pollution (Vadrevu et al., 2014; Cao, 2017). An important component of this pollution is the black carbon (BC) aerosol, the light-absorbing, refractory material produced mainly through the incomplete combustion of fossil fuels and biomass (Bond et al., 2013). In addition to its effects on air quality, BC plays a unique and important role in the Earth's climate system due to its impacts on solar radiation, clouds, and snow albedo (Bond et al., 2013). Several studies suggest that BC is the second largest contributor to anthropogenic radiative forcing after carbon dioxide due to its strong absorption of solar radiation (Jacobson, 2001; Ramanathan and Carmichael, 2008; Bond et al., 2013). Furthermore, BC aerosol can alter atmospheric circulation patterns, accelerate snowmelt, and cause glaciers to retreat (Xu et al., 2009).

Geographically, the TP is surrounded by South and East Asia where BC sources are strong (Zhang et al., 2009), and the TP has become impacted by these high-BC source areas due to the general circulation patterns (Cao et al., 2010; Lu et al., 2012; Zhao et al., 2017). For example, Lu et al. (2012) found that BC loadings in the Himalayas and TP increased by 41% from 1996 to 2010 due to the influence of surrounding areas, and on annual average South and East Asia accounted for 67% and 17% of BC transported to there, respectively. However, several recent studies showed that the impact of internal Tibetan sources (e.g., yak dung combustion by local residents) on the atmosphere of the TP should not be overlooked (Chen et al., 2015; Li et al., 2016a; Zhang X. et al., 2017). In the past few decades, a number of field campaigns conducted in the TP have investigated the concentrations, sources, and spatial and temporal variations of BC aerosol (e.g., Engling et al., 2011; Cong et al., 2015; Wang M. et al., 2016; Zhu et al., 2016; Zhao Z. et al., 2017). Recently, research has begun to focus on the light absorption characteristics of BC particles in the atmosphere and snow (Li et al., 2016b, 2016c; Zhang Y.et al., 2017), and these studies have been helpful for improving estimates of the radiative forcing of BC in the atmosphere of the TP.



Although some aerosol-related field studies have been conducted on the TP, the BC measurements were mainly made using online or offline filter-based techniques (e.g., aethalometer, thermal/optical reflectance method, and multi-angle absorption photometer) (e.g., Engling et al., 2010; Marinoni et al., 2010; Wan et al., 2015; Zhu et al., 2016; Li et al., 2017 ). For these techniques artifacts result from bulk particle deposition on the filters, and therefore, the direct determination of BC size and

mixing state with these techniques is infeasible. This is a limitation of the filter-based methods because the optical properties of BC aerosol are related to the particles' chemical and microphysical characteristics, including their size and mixing state. For instance, Liu et al. (2015) reported direct evidence of substantial field-measured BC absorption enhancement ($E_{abs}$) in an urban area where the magnitude of $E_{abs}$ was strongly dependent on internal mixing BC. Peng et al. (2016) used a novel environmental chamber approach to quantify the aging and variations in the morphology and optical properties of BC particles

from Beijing, China and Houston, United States. That study showed that BC particles initially changed from a fractal to spherical morphology with little change absorption followed by growth into compact particles with large $E_{abs}$.

Although accurate information on the physicochemical characteristics of BC is needed to improve our understanding of climate impacts on the TP, there is still lack of high-resolution measurements on the size and mixing state of BC in the region. This deficiency has led to considerable uncertainty in the calculations of BC direct radiative forcing over TP (He et al., 2014). In

this study, we used a single particle soot photometer (SP2) and a photoacoustic extinctiometer (PAX) to determine the mass concentrations, size distributions, mixing states, and light absorption properties of refractory BC (rBC) in the southeastern part of the TP. Various terms have been used in the literature for the most refractory and light-absorbing components of carbonaceous aerosols depending on the experimental measurement techniques (Bond et al., 2013). Here the term rBC is used exclusively in reference to SP2 measurements while BC more generally refers to measurements made with other techniques.

The primary objectives of this study were (1) to investigate the effects of meteorology on rBC and identify probable source regions responsible for the high rBC loadings; (2) to characterize the rBC size distributions and the evolution of rBC mixing state; and (3) to derive the rBC $E_{abs}$ and evaluate the factors that affect it.

## 2 Methodology

### 2.1 Sampling site

Physicochemical and optical properties of rBC aerosol were measured in samples from a remote area of Lulang, which is located on the southeastern TP (Fig. 1), and under the certain conditions samples air from the free troposphere. An intensive measurement campaign was conducted from 17 September to 31 October 2015 on the dormitory rooftop of the Integrated Observation and Research Station for Alpine Environment in South-East Tibet, Chinese Academy of Sciences (94.44 °E, 29.46 °N, ~3300 m above sea level). There were no major anthropogenic sources near the sampling site.



## 2.2 Data collection

### 2.2.1 Quantification of rBC mass, size, and mixing state

An SP2 manufactured by Droplet Measurement Technologies (Boulder, CO, USA) was used to determine the mass, size, and mixing state of rBC particles. The operation and principles of the SP2 have been described in detail elsewhere (Schwarz et al.,
2006). Briefly, a high-intensity intra-cavity Nd: YAG laser operating at wavelength of 1064 nm heats an individual rBC-containing particle to its incandescence temperature (~4000 K) which then emits thermal radiation that is detected by optical detectors. The intensity of the incandescence signal is proportional to the mass of rBC contained in the particle, and it is not affected by the particle morphology or the presence of non-refractory matter (Slowik et al., 2007). In this study, the SP2 was calibrated with a standard fullerene soot sample (Lot F12S011, Alfa Aesar, Inc., Ward Hill, MA, USA), and a linear
relationship was established between the peak intensity of the incandescence signal and the rBC mass. For this procedure, fullerene soot particles generated by an atomizer (Model 9302, TSI Inc., Shoreview, MN, USA) were passed through a diffusion silica-gel dryer, and then they were separated by size with a differential mobility analyzer (Model 3080, TSI Inc.) before entering the SP2 instrument. The corresponding fullerene soot masses were estimated using the effective density data provided by Gysel et al. (2011). More information concerning the SP2 calibration procedure may be found in Wang et al.
15  (2014).

The measured rBC mass was converted to the volume equivalent diameter (VED) by assuming rBC particles were solid spheres with a density of 1.8 g cm$^{-3}$ (Bond and Bergstrom, 2006). The detection efficiency of the SP2 dropped off for rBC core sizes <~70 nm or became saturated for sizes >~600 nm. Based on a mono-modal lognormal fit for the mass size distributions as described in section 3.3.1 below (Supplemental Fig. S1), the reported rBC mass concentrations in this study were scaled up by
a factor of ~1.1 to compensate the losses outside of the SP2 detection range. The uncertainty of the SP2 measurements was ~20%, which was estimated from the SP2 response to ambient rBC mass, sample flow, and estimates of the rBC mass beyond of SP2 detection range.

A major advantage of the SP2 is that it has the capability of determining the rBC mixing state (Schwarz et al., 2006). Freshly emitted rBC can be internally mixed with non-rBC materials through the process of gas-to-particle conversion. When the laser
beam in the SP2 heats an internally-mixed rBC particle, the coatings are preferential evaporated, and that causes a decrease in the intensity of the scattering signal. After that, the rBC core starts to vaporize, and that produces a peak in the incandescence signal. Therefore, there is a lag-time between the peaks of scattering and incandescence signals, and that lag-time can be used to characterize the internal mixing of rBC (McMeeking et al., 2011; Huang et al., 2012; Wu et al., 2016). Supplemental Fig. S2 shows that the lag-times exhibited a bimodal distribution with ~2 μs separating two distinct populations. The rBC-
containing particles with a lag-time >2 μs was considered to have substantial coatings, and those particles are denoted thickly-coated. In contrast, the rBC-containing particles with lag-times <2 μs were classified as uncoated or thinly-coated. Here the number fraction of thickly-coated rBC ($F_{rBC}$) was used to represent the degree of the internal mixing of rBC particles. It was calculated by dividing the number of thickly-coated rBC particles by the total number of rBC particles. Because there were no





incandescence signals detected for small particles and the scattering signal became saturated for large, coated rBC particles, the rBC core sizes used to evaluate internal mixing were limited to ~70 to 300 nm VED. The number size distribution of rBC shows that this was not a critical limitation in the following analysis because that size range contained the vast majority of the rBC particles (see Fig. S1).

### 2.2.2 Particle light absorption measurements

A PAX operating at wavelength of 870 nm ($PAX_{870}$, Droplet Measurement Technologies) was used to measure the particles' light absorption coefficients ($b_{abs}$) based on an intra-cavity photoacoustic technology. Sampled particles pass through a Nafion® dryer (MD-110-48S; Perma Pure, Inc., Lakewood, NJ, USA) before entering the $PAX_{870}$, and then light-absorbing particles are heated by the laser beam in the acoustic chamber. This heating produces a pressure wave that is detected with a

sensitive microphone. The $PAX_{870}$ can also measure the particles' light scattering coefficient ($b_{scat}$) simultaneously with a wide-angle integrating reciprocal nephelometer in the scattering chamber. Before and during sampling, the light scattering and absorption of the $PAX_{870}$ were calibrated with ammonium sulfate and freshly-generated propane soot, respectively. The light extinction coefficient ($b_{ext} = b_{scat} + b_{abs}$) can be calculated from the laser power of the $PAX_{870}$; thus a correction factor can be established from the relationship between the calculated $b_{abs}$ (= $b_{ext}$ - $b_{scat}$) and the measured $b_{abs}$. The $b_{ext}$ is calculated using

the following formula:

$$b_{ext} = -\frac{1}{0.354} \times \ln\frac{I}{I_0} \times 10^6 \ [Mm^{-1}] \tag{1}$$

where 0.354 is the path length of the laser beam through the cavity in meters; $10^6$ is a conversion factor used to express $b_{ext}$ in

$Mm^{-1}$; $I$ is the laser power during calibration, and $I_0$ is the average laser power before and after calibration. A linear relationship was established between the extinction-minus-scattering coefficients and the measured $b_{abs}$. The slope of the regression line, that is, the correction factor, was then used as the new calibration factor for absorption. In this study, the same steps for absorption calibration were repeated until the correction factor was stable within ~10%. The uncertainty of the PAX for absorption measurement was estimated to be ~15%. It is worth noting that the $b_{scat}$ produced by freshly-generated propane soot

particles has a substantial contribution to $b_{ext}$ while ammonium sulfate is the only material that generates $b_{scat}$. Thus, the scattering is calibrated before $b_{abs}$ using the same procedures as the absorption calibration.

### 2.2.3 Complementary data

A portable DustTrak aerosol monitor (Model 8530, TSI Inc., Shoreview, MN, USA) was used to measure the mass concentrations of total suspended particulate matter (TSP). Hourly ozone ($O_3$) was measured using a UV-based dual beam $O_3$

monitor (2B Technology model 205, CO, USA). Wind speed and wind direction were measured hourly with the use of an automatic weather station installed at the Integrated Observation and Research Station for Alpine Environment in South-East



Tibet, Chinese Academy of Sciences. The planetary boundary layer (PBL) depths were obtained from the National Centers for Environmental Predication (NCEP, http://apps.ecmwf.int/datasets). The spatial distribution of the BC column mass density was retrieved from the Modern-Era Retrospective analysis for Research and Applications version 2 (MERRA-2) using the Goddard Earth Observing System Model, Version 5 (GEOS-5) with its Atmospheric Data Assimilation System, version 5.12.4

(https://giovanni.gsfc.nasa.gov/giovanni). True color images obtained from the Moderate Resolution Imaging Spectroradiometer (MODIS) on the Terra satellite were used to assess the pollution distributions visually on several selected days, and these images can be downloaded from the website https://lance.modaps.eosdis.nasa.gov.

**2.3 Data analysis**

**2.3.1 Assessment of surface transport**

Hourly rBC concentrations and the corresponding wind data were used to estimate the surface transport of rBC at the Lulang site using the following formula (White et al., 1976):

$$f = \frac{1}{n}\sum_{j=1}^{n} C_j \times WS_j \times \cos\theta_j \tag{2}$$

where f is the surface transport intensity of rBC in units of $\mu g\ s^{-1}\ m^{-2}$ (that is, mass transported per unit time and area); $C_j$ and $WS_j$ are the mean rBC concentrations ($\mu g\ m^{-3}$) and wind speeds ($m\ s^{-1}$) during the *j*th observation hour, respectively; $\theta_j$ is the angle between wind direction and the north-south direction during the *j*th observation hour; and n is the total number of observation hours. In this study, we viewed the surface flux intensity as a measure of the influence of regional transport in the South Asia on the cross-boundary site using ground-based observations.

**2.3.2 Cluster analysis of air-mass trajectories**

Three-day air mass backward trajectories were used to characterize the transport pathways of rBC to Lulang. Each trajectory was calculated for an arrival height of 150 m above ground. The trajectories were calculated hourly using the Hybrid Single-Particle Lagrangian Integrated Trajectory (HYSPLIT) model (Draxler and Rolph, 2003) developed by the Air Resource Lab (ARL) in the National Oceanic and Atmospheric Administration (NOAA). Because a large number of trajectories (887)

retrieved for the entire campaign showed diverse pathways, a clustering procedure was used to determine representative pathways for the trajectories based on an angle-based distance statistics method. This was defined using the law of cosines (Sirois and Bottenheim, 1995) from the following equations:

$$d_{12} = \frac{1}{n}\sum_{i=1}^{n} \cos^{-1}\left(0.5 \times \frac{A_i + B_i - C_i}{\sqrt{A_i B_i}}\right) \tag{3}$$



$$A_i = (X_1(i) - X_0)^2 + (Y_1(i) - Y_0)^2 \tag{4}$$

$$B_i = (X_2(i) - X_0)^2 + (Y_2(i) - Y_0)^2 \tag{5}$$

$$C_i = (X_2(i) - X_1(i))^2 + (Y_2(i) - Y_1(i))^2 \tag{6}$$

where $d_{12}$ is the mean angle between the two backward trajectories, which varies between 0 and $\pi$; $X_0$ and $Y_0$ represent the position of the receptor site (Lulang in the present case); n is the total number of end points in a trajectory.

### 2.3.3 Concentration-weighted trajectory (CWT) model

A CWT model was used to construct the spatial distribution of the rBC sources that potentially influenced Lulang. For the CWT calculations, the entire geographic region covered by the three-day backward trajectories was separated into ~8100 grid cells of 0.5 °latitude ×0.5 °longitude. Each grid cell was assigned a residence-time weighted concentration obtained by hourly-averaged rBC concentration associated with the trajectories that crossed that grid cell (Hsu et al., 2003):

$$C_{ij} = \frac{\sum_{l=1}^{M} C_l \tau_{ijl}}{\sum_{l=1}^{M} \tau_{ijl}} \tag{7}$$

where $C_{ij}$ is the average weighted-concentration in the $ij$th grid cell; $C_l$ is the measured rBC concentration on the arrival of trajectory $l$; $\tau_{ijl}$ is the number of trajectory endpoints in the $ij$th grid cell by trajectory $l$; M is the total number of trajectories. A high $C_{ij}$ value indicates that air parcels travelling over the $ij$th grid cell would, on average, contribute to the high rBC loading at Lulang.

## 3 Results and discussion

### 3.1 Characteristics of surface rBC

#### 3.1.1 rBC loadings

A time-series plot of the hourly-averaged mass concentrations of rBC and TSP during the entire campaign is shown in Fig. 2. The hourly average mass concentrations of rBC ranged from 0.002 to 9.23 µg m$^{-3}$ with an arithmetic mean ($\pm$standard deviation, SD) of 0.31 ± 0.55 µg m$^{-3}$. A frequency distribution of the rBC mass concentrations (Supplemental Fig. S3) shows that the rBC values formed a typical truncated normal distribution, with ~60% of all the data below 0.2 µg m$^{-3}$. However, the coefficient of variation (defined as SD/mean) for rBC values was as high as 177%. Furthermore, ~25% of the rBC mass loadings were above the 75th percentile value of 0.33 µg m$^{-3}$. These results suggest that large loadings of rBC did at times occur at Lulang.





The grand average mass concentration of TSP for the study was $12.65 \pm 9.00$ μg m$^{-3}$, and TSP ranged from a minimum of 1.54 μg m$^{-3}$ to a maximum of 73.40 μg m$^{-3}$ (Fig. 2). The rBC particles accounted for 0.4–25.6% of TSP mass and averaged 2.6% of the TSP. Supplemental Fig. S4 shows that the relationship between rBC and TSP followed two different patterns. On 21 October, the mass concentrations of rBC were highly correlated with the TSP mass concentrations ($r = 0.97$), but a weaker

correlation ($r = 0.67$) was found for the other sampling days. Moreover, rBC accounted for 13.6% of TSP mass on 21 October, but the contribution was considerably smaller (2.2%) for other sampling days. As rBC is produced by combustion (Bond et al., 2013), these results indicate that combustion sources contributed significantly to TSP mass on 21 October while particles from non-combustion related sources, such as secondary aerosols and soil dust, were relatively more abundant on the other sampling days.

Fig. 1 shows the spatial distribution of BC mass concentrations at different high altitude locations on the Himalayas and TP, and information for each study from which results were taken is summarized in Table S1. Although the sampling periods differed among the studies, BC generally exhibited larger loadings in the Himalayan foothills compared with those observed on the TP. On the other hand, the BC mass concentrations varied inversely with the altitude of the sampling sites ($r = 0.81$) (Fig. 1). The average rBC mass concentration at Lulang was higher than what has been measured in the interior or northern

TP, but it was lower than at several locations on the southeastern TP and the Himalayan foothills (Fig. 1). The differences in BC loadings among locations can be attributed to several factors; first, the concentrations are affected by pollutant transport from upwind regions (e.g., South Asia), and that transport is affected by the complex topography of the area. For example, Zhang et al. (2015) found that on annual average ~50% of the BC column burden of the Himalayas and TP was due to transport from South Asia (~33% by biomass and biofuel emissions and ~17% by fossil fuel emissions). Meanwhile, Zhao S. et al.

(2017) concluded that the high altitudes of the Himalayas and TP can impede the transport of BC from the Indo-Gangetic Plain (IGP) to these regions, and therefore, the trans-Himalaya transport of BC is strongly dependent upon meteorological conditions over the IGP. Second, the uncertainties caused by the inherent limitations of instruments themselves also help explain the variability in loadings. Indeed, previous studies have shown that the BC obtained from filter-based optical techniques (e.g., aethalometer) is affected by the light scattering artifacts (Virkkula et al., 2007) while laser-induced incandescence methods

(e.g., SP2) can undersample small particles (Bond et al., 2013). Finally, there is still a lack of BC method intercomparisons, and the differences among methods may be greater in remote areas than urban ones. For example, Wang et al. (2014) reported that a scaling factor of 2.5 was needed to correct the BC mass concentrations measured with an aethalometer to match SP2 measurements at a remote site on the northeastern TP while the corresponding value at an urban site was 1.3. Moreover, filter-based BC measurements based on thermal-optical reflectance methods may be affected by the presence of carbonates (Li et

al., 2017), and mineral dust particles, including carbonates, can contribute considerably to the aerosol populations in some areas of the TP due to the general lack of vegetative cover. Currently, there are still difficulties in obtaining scaling factors to reconcile the various BC measurements on the TP to a common standard, and therefore, direct comparisons of BC data obtained by different methods can be tenuous.



### 3.1.2 Diurnal variations

Fig. 3 (a–c) shows the diurnal variations of rBC mass concentrations, PBL depths, and wind speeds over the course of the campaign. The rBC mass concentrations decreased slightly after midnight to reach a low value of 0.16 μg m$^{-3}$ in the early morning, around 05:00 local time (LT—all time references below are given in LT); that was followed by a sharp increase at a rate of 0.35 μg m$^{-3}$ h$^{-1}$ to a maximum value of 1.21 μg m$^{-3}$ around 09:00. The rBC loadings then decreased rapidly at a rate of 0.36 μg m$^{-3}$ h$^{-1}$ and subsequently reached a diurnal minimum of 0.10 μg m$^{-3}$ in the afternoon around 14:00. Thereafter, the rBC again increased gradually to a small peak of 0.26 μg m$^{-3}$ at night around 20:00. After that, the concentrations were relatively stable until 01:00.

Previous studies in urban areas have often shown a morning peak in BC/rBC linked to local rush hour traffic (e.g, Cao et al., 2009; Wang et al., 2016a). In contrast, slight morning enhancements in BC/rBC have been found at some sites on the TP, and they were attributed to local anthropogenic activities (e.g., Wang et al., 2014; Wang M. et al., 2016). In our study, a morning peak also was observed at Lulang, but the rBC loadings enhancements were as large as six-fold over minimum values. The morning peak was consistent with the activities of the local residents, especially cooking, indicating that there were some contributions of rBC from local sources. However, local emissions alone may not explain such a large difference in concentrations. This can be assessed indirectly by comparisons of the morning peaks with the much smaller rBC enhancements in the evening around 19:00–20:00, which also are influenced by local cooking activities. Thus, the large morning peaks may result from the combined effects of local activities and regional transport. As shown in Fig. 3 (a–b), the rapid morning increase in rBC was accompanied deepening of the PBL, and therefore, regional transport maybe an important influence on the aerosol populations. Located to the southwest of Lulang, Bangladesh and the IGP are known to be strong sources for BC particles (Zhang et al., 2009). The PBL is typically shallow and stable at night, and pollutants from the IGP and Bangladesh tend to be confined near the surface at that time. After sunrise, as the PBL starts to deepen, strengthening thermals lift and eventually break the nighttime inversion, and this can lead to the transport of pollutants to the southeastern TP.

The preceding explanation concerning the effects of transport is supported by the analysis of true color images of haze clouds retrieved by the MODIS on the Terra satellite (Supplemental Fig. S5). That satellite passes over the study region at ~10:30, and even though only several sampling days (20–23 October) were selected for inclusion in Fig. S5, most days exhibited similar patterns. The true color images reveal obvious pollution bands along the IGP and Bangladesh, which piled up on the south margin of the TP. The prevailing wind direction around the southeastern margin of the TP was southerly (Fig. S5), and the pollutants in the pollution bands could be transported to the sampling site along the valley of the Yarlung Tsangpo River; indeed, this has been considered as a "leaking wall" for pollutant transport to the southeastern TP (Cao et al., 2010). The decreasing trend in the late morning at Lulang can be explained by the continued deepening of the PBL and increased wind speeds (see Fig. 3 b and c), and those meteorological conditions also can explain the daily minima in the rBC loadings in the afternoon because they cause the dilution and dispersal of the aerosol. The slight enhancement of rBC at night can be attributed



to shallow PBLs and low winds in addition to local increased rBC anthropogenic emissions from daily activities, such as cooking and heating.

## 3.2 Meteorological effects on rBC concentrations

Wet deposition was found to be the major removal mechanism for BC aerosol (Bond et al., 2013). During the rain events at
Lulang, the hourly precipitation varied from 0.2 to 4.0 mm (Fig. 2), and the total precipitation during the campaign was 104.8 mm. Rain events occurred on ~30% of the sampling period, and ~70% of the rain occurred in September due to the influx of moist warm air from the Indian and Pacific Oceans (Kang et al., 2002). The average mass concentration of rBC during rainy days (0.25 ± 0.13 µg m$^{-3}$) was ~45% lower than on non-rainy days (0.36 ± 0.38 µg m$^{-3}$). Supplemental Fig. S6 shows the impact of daily precipitation on rBC loadings; that is, the rBC mass concentrations were negatively correlated with
precipitation (r = -0.51). In the classification scheme for daily precipitation issued by the China Meteorological Administration (GB/T 28592–2012), light, moderate, and heavy rain are defined as precipitation ranges of 0.1–9.9, 10.0–24.9, and 25.0–49.9 mm within 24 h, respectively. When the daily precipitation was less than 10 mm, the rBC loadings had large fluctuations, ranging from 0.06 to 0.45 µg m$^{-3}$. However, when the daily precipitation was higher than 10 mm, the rBC values were < 0.14 µg m$^{-3}$, suggesting that rBC particles are removed more efficiently by moderate or strong rain compared with light rain.
Wind speed and wind direction play crucial roles in the dilution and transport of pollutants (Fast et al., 2007), and Fig. 4a shows the wind speeds and directions during the study. Overall, the prevailing surface wind directions were westerly and northerly, and these sectors combined accounted for ~70% of the total wind frequencies. The average wind speed was 1.07 ± 0.93 m s$^{-1}$, and the higher wind speeds were most often associated northerly flow.

To investigate possible effects of horizontal advection of rBC, we examined the relationships between rBC loadings and wind
speed and wind direction using a bivariate polar plot (Fig. 4b). When the wind speed exceeded 1 m s$^{-1}$, large rBC loadings were associated with airflow from the southeast, and that is the compass sector for transport from Yarlung Tsangpo River Valley, which as noted above can bring pollutants from the IGP and Bangladesh to our site (Cao et al., 2010). High rBC mass concentrations also occurred under static conditions or low winds (<1 m s$^{-1}$), which typically promote the accumulation local pollutants near the Earth's surface. In contrast, low levels of rBC were observed when the winds were from the north-northwest,
and this is likely because upwind regions in those directions contain few rBC sources, and therefore strong wind speeds from the north-northwest sectors tend to dissipate the rBC particles.

To evaluate the surface transport of rBC to Lulang from the south (arbitrarily designated as positive, from outside the TP, e.g., IGP and Bangladesh) and north (negative, from the interior of the TP), surface transport intensities were calculated from Eq. (2) based on the observed rBC mass concentrations, wind speed, and wind direction at the sampling site. The estimated overall
net surface transport of rBC was +0.05 ± 0.29 µg s$^{-1}$, indicating greater transport of rBC from outside of the TP than from the interior of the TP. The large coefficient of variation (580%) of the surface transport intensity reflects strong fluctuations in the transport, and two factors likely influenced the transport processes. First, the surface fluxes were more than likely strongly affected by the prevailing winds. Fig. 5 shows the variations in the hourly-averaged surface transport intensity of rBC and the





corresponding wind vectors (m s$^{-1}$). In general, the rBC transport intensities exhibited a clear "saw-toothed" pattern, with changes in the influx (positive) and outflux (negative) patterns corresponding to shifts in wind direction (Fig. 5a). Second, differences in the emission intensities for pollutants in the upwind areas are another factor that likely affected the transport of rBC. For example, as noted previously, the emissions of rBC from the IGP and Bangladesh are strong, and the average rBC

influx intensity ($+0.18 \pm 0.27$ μg s$^{-1}$ m$^{-2}$), which includes transport from these areas, was two-fold stronger than the efflux intensity ($-0.09 \pm 0.24$ μg s$^{-1}$ m$^{-2}$) (Fig. 5b).

### 3.3 Effects of regional transport

Fig. 6a shows the three cluster-mean trajectories that were calculated from the individual three-day backward trajectories for the campaign. For discussion purposes, we arbitrarily defined a trajectory as "polluted" if it corresponded to an rBC

concentration higher than the 75th percentile value of 0.33 μg m$^{-3}$; otherwise it was classified as "clean" trajectory. The average rBC mass concentrations for the three clusters and the polluted trajectories are summarized in Table 1. The air masses grouped into Cluster #1 originated from north India, and the air masses grouped into this cluster passed through central Nepal and the southern TP before arriving at Lulang. The average rBC mass concentration for Cluster #1 was $0.37 \pm 0.71$ μg m$^{-3}$. Of all 887 backward trajectories included in the analysis, ~47% were allocated to Cluster #1, and ~29% of those were considered polluted.

The average rBC mass concentration for these polluted trajectories was 0.95 μg m$^{-3}$, and the air masses grouped into Cluster #1 were responsible for many of the high rBC loadings at the receptor site. The air masses associated with Cluster #2 originated from central Bangladesh and then moved across northeastern India and to the southeast of Tibet before arriving at Lulang. The average rBC mass concentration for Cluster #2 was $0.24 \pm 0.36$ μg m$^{-3}$. The percentage of the trajectories assigned to this cluster was ~44%; ~20% of those were regarded as polluted; and the average mass concentration of the polluted trajectories

in Cluster #2 was 0.75 μg m$^{-3}$. The air masses in Cluster #3 originated over central Tibet. The average rBC mass concentration for this cluster ($0.32 \pm 0.31$ μg m$^{-3}$) was similar to that for Cluster #1. Although the percent contribution from this cluster was ~9% of all trajectories, ~30% of the trajectories in Cluster #3 were classified as polluted and had a mean value of 0.72 μg m$^{-3}$; this implies some contributions of rBC from internal Tibetan sources.

A CWT model was used to identify the locations of the potential source areas that provided rBC to Lulang. Fig. 6b shows a

map of CWT results for the campaign, and there were three main source regions contributing to the rBC pollution at Lulang. Region I was mainly composed of areas along the southern border of the Himalayan foothills, IGP, and north Bangladesh. This region had the highest CWT values, indicating that these areas had the greatest probability for transporting high rBC to Lulang. Further evidence of this can be seen in the map of average BC column mass densities in the TP and South Asia during the sampling period (Fig. 6c)—it confirms large BC loadings along the southern border of Himalayan foothills, IGP, and

Bangladesh, and this is consistent with the large CWT values in Region I discussed above. Although the high altitude of the Himalayas can inhibit air flow from Region I to the TP, previous studies indicate that pollutants can be transported over the Himalayas to southeast TP through some valleys, including that the Yarlung Tsangpo Valley (Cao et al., 2010; Zhao S. et al., 2017).





In contrast, moderate CWT values were found in areas to the west of Lulang and adjoining regions (Region II), suggesting local anthropogenic activities in the interior of the TP also contributed to the rBC loadings at Lulang. Several cities, including Lhasa, Gongbu Jiang and Linzhi, are located ~60–350 km to the west of Lulang, and these are presumptive sources for anthropogenic materials. Thus, air masses that arrived at Lulang from the west may have carried pollutants to the sampling site. Although the population is sparse in the areas surrounding Lulang, biofuels (e.g., wood and yak dung) are the main energy sources for the local residents (Ping et al., 2011). Emissions from heating and cooking contain large quantities of rBC particles, and hence these domestic sources probably affected the sampling site. Region II evidently had less affects effects on the rBC loadings compared with the Region I because the CWT values for Region II were lower. It's worth noting that although Region III extended to the southwest of Sinkiang Province, China and several central Asian countries, this source had only minor impacts on the rBC because the air masses from Region III composed less than ~1% of the total trajectories.

### 3.4 Microphysical properties

### 3.4.1 Size distributions of rBC

Fig. S1 shows that rBC core size distribution was well represented by a mono-modal lognormal fit, which is consistent with previous SP2-based observations made across the globe, including urban, rural, remote areas (e.g., Schwarz et al., 2008; Liu et al., 2010; McMeeking et al., 2011; Huang et al., 2012; Wang et al., 2014). As shown in Fig. 2, the hourly-averaged mass median diameters (MMD—the VED at the peak of the mass distribution) varied broadly from 98 to 255 nm during the study and the average of these was 160 ±23 nm. The rBC MMDs exhibited diurnal patterns similar to the rBC mass concentrations; that is they peaked in the morning around 09:00 (~183 nm), fell to a minimum in the afternoon around 14:00 (~147 nm), then rose again in the evening, and finally stabilized at night (~163 nm) (Fig. 3d).

Although size-segregated filter-based measurements made with cascade impactors provide information on the aerodynamic diameters of BC particles, they measure both the BC cores and any coatings. In contrast, SP2 measures the rBC core size alone. Consequently, we only compared our results with SP2 observations made in previous studies. Because of the different rBC densities assumed in the various studies, we normalized them to the same density of 1.8 g cm$^{-3}$ to facilitate direct comparisons. Although the average rBC MMD at Lulang fell into the lower range reported in previous SP2 studies (~155–240 nm; Huang et al., 2012, and references therein), it was lower than some results reported for remote areas, such as 181 nm at Qinghai Lake, northeastern TP (Wang et al., 2014), 194 nm at the Pallas Global Atmosphere Watch station, Finnish Arctic (Raatikainen et al., 2015), and 220–240 at the high alpine research station Jungfraujoch, Switzerland (Liu et al., 2010).

The variations in rBC MMDs among different sites are likely related to the following factors. First, the various emission sources produce rBC particles of different sizes. For example, Sahu et al. (2012) observed larger average rBC MMD in biomass burning plumes (193 nm) than that in fossil fuel plumes (175 nm), and Wang et al. (2016b) reported a higher average rBC MMD for coal burning (215 nm) compared with particles from a traffic source (189 nm). Second, air mass transport histories affect the size distributions of rBC. Take the cluster analysis as an example: the average rBC MMD was the largest (184 nm)



when the polluted air masses originated from central Bangladesh (Cluster #2). In contrast, smaller rBC MMDs were found when the polluted air masses came from North India (Cluster #1, 173 nm) or central TP (Cluster #3, 177 nm). Moreover, more aged particles in the plumes tend to be larger than fresher particles from close to the source (Moteki et al., 2007). Finally, wet deposition may exert a significant impact on the rBC size distributions. This can be seen in Fig. 7 which presents a comparison

of the frequency distributions of rBC MMDs during rainy and non-rainy sampling days. The rBC MMDs varied from 112 to 255 nm with an average of $164 \pm 21$ nm for the non-rainy days, and ~50% of the MMDs were within the range of 150–175 nm. In contrast, the rBC MMDs for rainy days shifted toward small sizes, varying from 98 to 230 nm and averaging $145 \pm 25$ nm. About 40% of the MMDs for the rainy day samples were in the range of 125–145 nm. The size on rainy days may be representative of local sources because rain also fell over South Asia, and therefore, there was little long range transport of

rBC to Lulang. Compared with the non-rainy days, the smaller rBC size on rainy days can be explained by the reduced atmospheric lifetimes for large rBC particles due to wet scavenging.

### 3.4.2 Evolution of rBC mixing state

The hourly-averaged $F_{rBC}$ ranged from 20 to 68% (average = $39 \pm 8$%, Fig. 2), and that is lower than what has been reported for Qinghai Lake (59%, Wang et al., 2015a) where a similar method was used to measure the internal mixing of rBC. Air

masses in Cluster #2 showed the highest internal mixing of rBC particles (40%), followed by Cluster # 1 (38%), and Cluster #3 (34%). The low percentages of internal mixing for rBC particles in these three clusters indicates a relatively low level of particle aging, and this implies that freshly-emitted local rBC particles may have been part of the sample population. Fig. 8a shows that the diurnal cycle of $F_{rBC}$ at Lulang typically exhibited "two peaks and two valleys". The internally-mixed rBC reached a peak value of 45% in the morning around 07:00–08:00, followed by a decreasing trend to a low value of 35% around

10:00. The internally-mixed rBC then increased to a secondary peak value of 44% in the afternoon around 14:00, and again slowly decreased to a minimum value of 33% around 01:00.

The variations of internally-mixed rBC in the morning further provide evidence for the combined effects of local activities and regional transport on the rBC aerosol. That is, the enhancement of internally-mixed rBC around 07:00–08:00 can be attributed to rBC aging, which indicates impacts from regional transport. The decreasing trend of $F_{rBC}$ around 09:00–10:00 was likely

due to an increase in fresh rBC particles emitted by local anthropogenic activities such as cooking even though the local population was small. As the day progressed, between 10:00 and 19:00, $F_{rBC}$ varied with $O_3$ mixing ratios (Fig. 8a). To further investigate the impact of oxidant levels on the internal mixing of rBC, the $F_{rBC}$ collected between 10:00–19:00 was plotted against the $O_3$ mixing ratios. It can be seen in Fig. 8b that $F_{rBC}$ was positively correlated with the $O_3$ mixing ratios (r = 0.89), indicating that more internal mixing for rBC particles occurred under more oxidizing conditions. Further, the observed

increasing trend for internally-mixed rBC from 10:00–14:00 can be explained by mixing of secondary aerosols (e.g., non-refractory inorganic and organic compounds) with the rBC particles due to enhanced photochemical oxidation. As shown in Fig. 8b, the slope of $F_{rBC}$ versus $O_3$ during midday was 0.44 % ppb$^{-1}$, and this reflects the rate at which oxidation affects rBC





mixing. In contrast, the decrease in $F_{rBC}$ from 15:00–20:00 can be ascribed to the small impacts from secondary aerosols due to less oxidizing conditions.

To further investigate the effects of photochemical oxidation on the rBC mixing state, we compared the diurnal variations of internal mixing for rBC particles at Lulang and Qinghai Lake, a site in the northeastern TP where studies were conducted in

October 2011 (Fig. 8c). The rBC and $O_3$ at Qinghai Lake were measured with the same type of SP2 as in this study and an ultraviolet photometer, respectively, and detailed descriptions of the Qinghai Lake study may be found in Wang et al., 2014; 2015b. As shown in Fig. 8c, only one $F_{rBC}$ peak was observed at Qinghai Lake in the afternoon between 12:00–17:00, and that was different from what we observed at Lulang. This difference can be explained by the fact that rBC in the early morning at Qinghai Lake was not affected by long-range transport owing to the topography of the region (Wang et al., 2014). Even so,

similar to Lulang, the $F_{rBC}$ during the daytime (08:00–18:00) at Qinghai Lake was positively correlated with the $O_3$ mixing ratio (r = 0.75, Fig. 8d). However, at Qinghai Lake, the slope of the $F_{rBC}$ versus $O_3$ was 0.24% ppb$^{-1}$ from 08:00–16:00, and therefore, the rate at which the internally-mixed rBC increased with $O_3$ was lower than at Lulang by a factor of ~2. The more rapid increase in $O_3$ at Lulang (4.8 ppb h$^{-1}$) compared with Qinghai Lake (3.2 ppb h$^{-1}$) shows that the oxidation rates leading to the formation of internally mixed rBC at Qinghai Lake were lower than at Lulang even though $O_3$ was higher at Qinghai

Lake (58.7 ppb) than that at Lulang (32.1 ppb). Similar results have been found in urban areas of Beijing and Xi'an, China (Wang et al., 2017), and the relevant conclusion from that study was that photochemical oxidation can be important for formation the coatings on rBC particles.

**3.5 rBC optical properties**

Hourly-averaged $b_{abs}$ at $\lambda$ = 870 nm measured with the PAX$_{870}$ varied from 0.1–22.2 Mm$^{-1}$ with a grand average for the study

of 2.1 $\pm$ 2.0 Mm$^{-1}$ (Fig. 2). Some organic materials (also called brown carbon) can cause significant light absorption but only in the short wavelengths (e.g., $\lambda$ = 370 nm) and they have nearly no absorption in the near-infrared spectral region (e.g., $\lambda$ = 870 nm) (Laskin et al., 2015). Consequently, the mass absorption cross section of rBC (MAC$_{rBC}$, m$^2$ g$^{-1}$), which describes the degree of light absorption per-unit-mass of rBC, can be calculated by dividing the $b_{abs}$ measured with the PAX$_{870}$ by the mass concentration of rBC detected with the SP2 (MAC$_{rBC}$ = $b_{abs}$/rBC). Fig. 9 shows that the MAC$_{rBC}$ frequency distributions for

all samples from the campaign and for the data stratified by the three trajectory clusters all were mono-modal and lognormals. The peak in the frequency MAC$_{rBC}$ distribution for the entire campaign was 7.0 m$^2$ g$^{-1}$, and there were slightly higher values for Cluster #1 (7.2 m$^2$ g$^{-1}$) and Cluster #2 (7.4 m$^2$ g$^{-1}$) compared with Cluster #3 (6.7 m$^2$ g$^{-1}$).

The rBC absorption enhancements ($E_{abs}$ = MAC$_{rBC}$/MAC$_{rBC,uncoated}$) were calculated to further investigate the rBC particles' optical properties. As the SP2 only determines the rBC core size, the hourly-averaged MMDs for the rBC were input into the

Mie model to calculate the MAC$_{rBC}$ of uncoated rBC particles (MAC$_{rBC,uncoated}$), with the assumption that the uncoated rBC particles were spherical and homogeneous. A more detailed description of the Mie algorithms may be found in Bohren and Huffman (2008). For these calculations, the refractive index of 1.85-0.71$i$ at $\lambda$ = 550 nm suggested by Bond and Bergstrom (2006) was first used in the Mie model to estimate the MAC$_{rBC,uncoated}$. Those values were then converted to the MAC$_{rBC,uncoated}$




at $\lambda$ = 870 nm based on an rBC absorption Ångström exponent of 1.0 (Moosmüller et al., 2011). Finally, the average rBC absorption enhancement was calculated by comparing the $MAC_{rBC}$ at $\lambda$ = 870 nm for rBC with and without coatings. As shown in Fig. 9, there were several extremely large anomalous $MAC_{rBC}$ values that were likely caused by the uncertainties due to extremely low $b_{abs}$ and rBC mass concentrations. To avoid spurious results, only $MAC_{rBC}$ values lower than the 90th percentile of all data were used to calculate the $E_{abs}$. As shown in Supplemental Fig. S7, the $E_{abs}$ values generally followed a mono-modal lognormal distribution with a peak value of 1.8, which is an indication that the light absorption by coated rBC particles was significantly greater than that by uncoated ones.

To investigate the potential impacts of rBC size and mixing state on light absorption, the $E_{abs}$ values were plotted against the $F_{rBC}$ values and MMDs (Fig. 10). As shown in Fig. 10a, the $E_{abs}$ was strongly positive correlated with the $F_{rBC}$ (r = 0.96), and this supports our conclusion that there was an enhancement of light absorption by internally-mixed—that is, coated—rBC particles. The slope of the regression line was 0.03 %$^{-1}$, and this may be considered a rough estimate of the effects of the coatings on light absorption. This means that if the internally-mixed rBC particles increased by one percent, the rBC particles would absorb 3% more light. If the results of the linear regression shown in Fig. 10a are extrapolated to a condition in which rBC is completely uncoated (that is, $F_{rBC}$ or x = 0%), the $E_{abs}$ would be 1.1, which is close to the theoretical value of 1.0 for uncoated rBC. At the other extreme, if all rBC particles were internally mixed ($F_{rBC}$ or x = 100%), the $E_{abs}$ would be as high as 4.3, which appears physically implausible. This result is confined to a narrow range of conditions, however, that is, small rBC core diameters with the thick coatings (Bond et al., 2006). Moreover, it is noteworthy that several studies have shown non-linear relationships between $E_{abs}$ and the internal mixing of rBC (e.g., Zhang et al., 2016; Liu et al., 2015), and in those cases, the $E_{abs}$ tended to be stable over a large range of coating thicknesses. If that is the case, the $E_{abs}$ should be lower than the calculated value of 4.3.

As shown in Fig. 10b, the $E_{abs}$ was non-linearly related to the MMDs of the rBC. When rBC MMD <170 nm, the $E_{abs}$ correlated negatively with the rBC core size, indicating that smaller rBC particles potentially have a stronger ability to amplify light absorption than large ones. This can be explained by the small rBC particles greater tendency to form coatings than the large ones (see the positive correlation between $F_{rBC}$ and MMDs in Supplemental Fig. S8) due to the well-known relationship between particle surface area and volume. The variations in $E_{abs}$ were relatively constant for rBC MMD >170 nm. This is because that larger rBC cores have smaller degree of internal mixing and weaker absorption amplification on the one hand. But on the other hand, larger rBC core size also would decrease the $MAC_{rBC,uncoated}$ calculated by the Mie model (see the relationship between $MAC_{rBC,uncoated}$ and MMD in Fig. S8). Eventually, the decrease in both the light absorbing ability for the measured ambient rBC (that is, the $MAC_{rBC}$) and for the assumed uncoated rBC particles (that is, $MAC_{rBC,uncoated}$) would cancel out causing a constant value for $E_{abs}$. Bond et al. (2006) similarly reported that the amplification was nearly constant for rBC cores >~150 nm.





## 4 Conclusions

The mass concentrations, size distributions, mixing state, and optical properties of rBC aerosol were studied at Lulang on the southeastern TP, China. The mass concentration of rBC, averaged over the entire campaign, was $0.31 \pm 0.55$ µg m$^{-3}$, and those particles accounted for 2.6% of TSP mass. A clear diurnal pattern in rBC mass concentrations was observed: high values

occurred in the early morning due to the combined effects of local anthropogenic activities and regional transport while low values in the afternoon were ascribed to dissipation of the rBC due to deepening of the PBL and higher wind speeds. The relationship observed between rainfall and rBC indicated that rBC particles are more efficiently removed by moderate and heavy precipitation (>10 mm) than by light rain. A bivariate polar plot showed that high rBC loadings were associated with strong winds from the southeast or static wind conditions. The estimated overall net surface transport intensity of rBC was

$+0.05 \pm 0.29$ µg s$^{-1}$ m$^{-2}$, and those calculations showed that more rBC was brought to the site from outside the TP than from interior of the TP. Moreover, air mass trajectory clusters and a concentration-weighted trajectory model indicated that sources in north India were the most important influences on rBC at Lulang, but local contributions were not negligible.

The rBC VEDs showed approximately mono-modal lognormal distributions. The hourly-averaged rBC MMD was $160 \pm 23$ nm, and the particle sizes varied among different air parcels. The MMDs shifted toward smaller sizes (145 nm) on rainy days

compared with the non-rainy days (164 nm). The average $F_{rBC}$ for the study was $39 \pm 8$%, suggesting uncoated or thinly-coated rBC particles composed the bulk of the rBC aerosol. Two peaks in $F_{rBC}$ were observed: one was in the morning, and that was attributed to atmospheric aging processes; the other was in the afternoon, and that was explained by enhancements caused by photochemical oxidation. A strong correlation between $F_{rBC}$ and $O_3$ was found during the daytime at Lulang (10:00–19:00), indicating that the photochemical oxidation plays an important role in the internal mixing of rBC with other materials. A

similar relationship was found for samples from Qinghai Lake in the northeastern TP, but regression analyses showed that the rate of increase in $F_{rBC}$ with $O_3$ was greater at Lulang than at Qinghai Lake (0.44 versus 0.24% ppb$^{-1}$). These results suggest that the rBC particles in Lulang formed internal mixtures more rapidly than those at Qinghai Lake.

Hourly-averaged $b_{abs}$ (at $\lambda = 870$ nm) varied from 0.1–22.2 Mm$^{-1}$ over the study with an overall average of $2.1 \pm 2.0$ Mm$^{-1}$. The $MAC_{rBC}$ values showed a mono-modal lognormal distribution, and a peak value of 7.0 m$^2$ g$^{-1}$. Slightly higher $MAC_{rBC}$

values were found for air masses from North India (7.2 m$^2$ g$^{-1}$) and central Bangladesh (7.4 m$^2$ g$^{-1}$) compared with air transported from central Tibet (6.7 m$^2$ g$^{-1}$). By dividing the observed $MAC_{rBC}$ measured with the SP2 and PAX$_{870}$ by the $MAC_{rBC,uncoated}$ calculated from the Mie model, the average $E_{abs}$ was estimated to be 1.8, suggesting that the light absorption by coated rBC particles was significantly amplified compared with the uncoated ones. Furthermore, the $E_{abs}$ was positively correlated with $F_{rBC}$, indicating an enhancement of light absorption by internally-mixed rBC particles. The $E_{abs}$ showed a

negative correlation with the rBC MMDs for the particle cores <170 nm, but it was nearly constant for larger rBC cores.



## Acknowledgments

This work was supported by the National Natural Science Foundation of China (41230641, 41503118, 41625015, and 41661144020). The authors are grateful to the Integrated Observation and Research Station for Alpine Environment in South-East Tibet, Chinese Academy of Sciences, for their assistance with field sampling.

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



**Table 1.** Trajectory clusters and mean rBC concentration for each cluster.

| Cluster | All trajectories | | | Polluted rBC trajectories[a] | | |
|---|---|---|---|---|---|---|
| | Number | Mean | SD[b] | Number | Mean | SD |
| #1 | 421 | 0.37 | 0.71 | 120 | 0.95 | 1.14 |
| #2 | 390 | 0.24 | 0.36 | 81 | 0.75 | 0.52 |
| #3 | 76 | 0.32 | 0.31 | 23 | 0.72 | 0.29 |
| All | 887 | 0.31 | 0.56 | 224 | 0.86 | 0.90 |

[a]Trajectories associated with rBC concentration >0.33 µg m$^{-3}$ (75th percentile value).

[b]SD represents standard deviation.



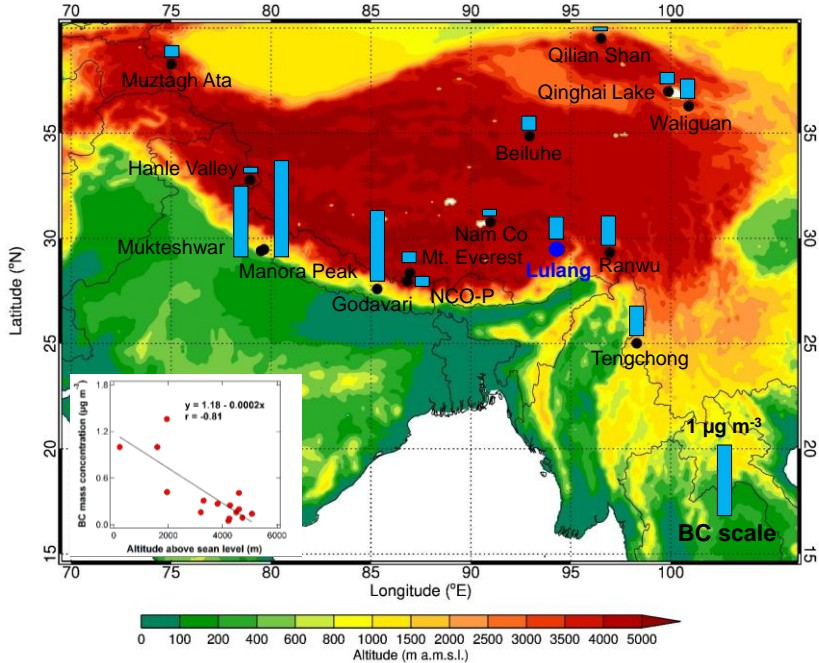

**Figure 1.** Black carbon concentrations (μg m$^{-3}$) measured at 15 sampling sites on Himalayas and Tibetan Plateau based on the measurements from this study (blue solid circles) and other studies (black solid circles) from Ma et al. (2003), Pant et al., 2006, Marinoni et al. (2010), Stone et al. (2010), Babu et al. (2011), Engling et al. (2011), Zhao et al. (2012), Li et al. (2017), Wan et al. (2015), Wang et al. (2015a), Wang M. et al. (2016), Zhu et al. (2016), and Raatikainen et al. (2017). More detailed information concerning these studies is summarized in Table S1. The lower-left corner panel is a scatter plot of the mass concentrations of BC versus the altitude of each sampling site. The map in the figure was drawn by the Weather Research and Forecasting (WRF) model.





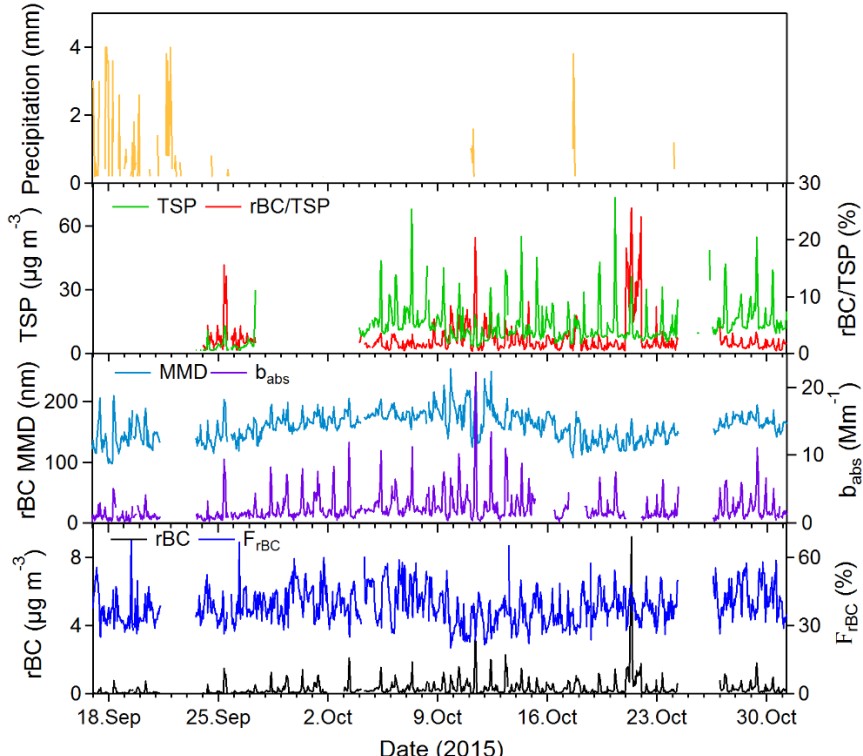

**Figure 2.** Time series of hourly-averaged rBC mass concentrations, number fraction of thickly-coated rBC ($F_{rBC}$), mass median diameter of rBC particles (MMD), total suspended particulate matter (TSP), rBC/TSP, light absorption coefficient ($b_{abs}$), and precipitation.



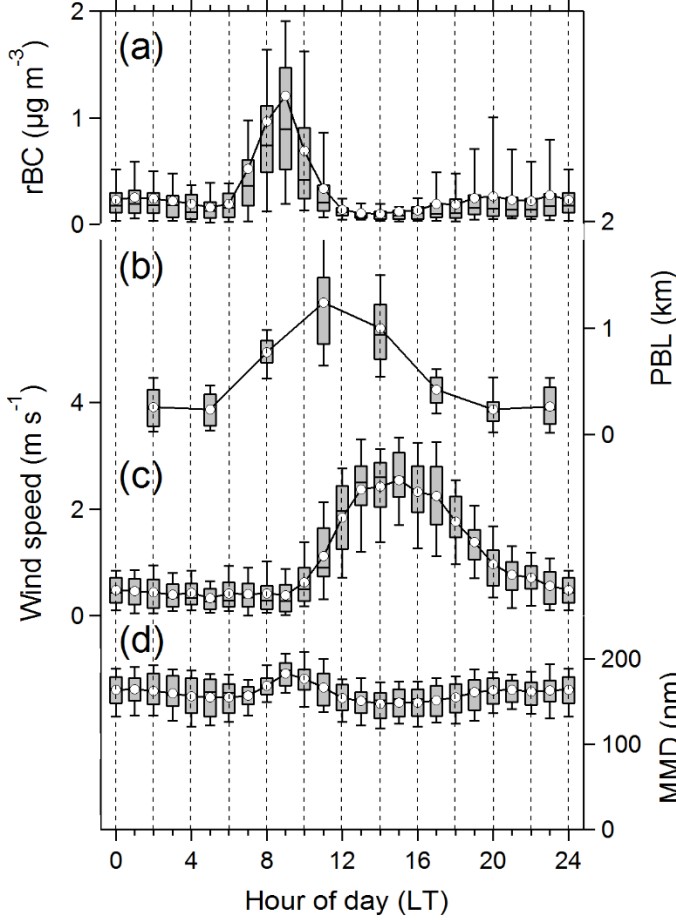

**Figure 3.** Diurnal variations of (a) rBC mass concentration, (b) planetary boundary layer (PBL) depth, (c) wind speed, and (d) mass median diameter of the rBC particles (MMD). The lower and upper edges of the boxes denote the 25% and 75% percentiles, respectively. The short black lines and white circles inside the boxes indicate the median and mean values, and the vertical bars ("whiskers") show the 10th and 90th percentiles. LT stands for local time.


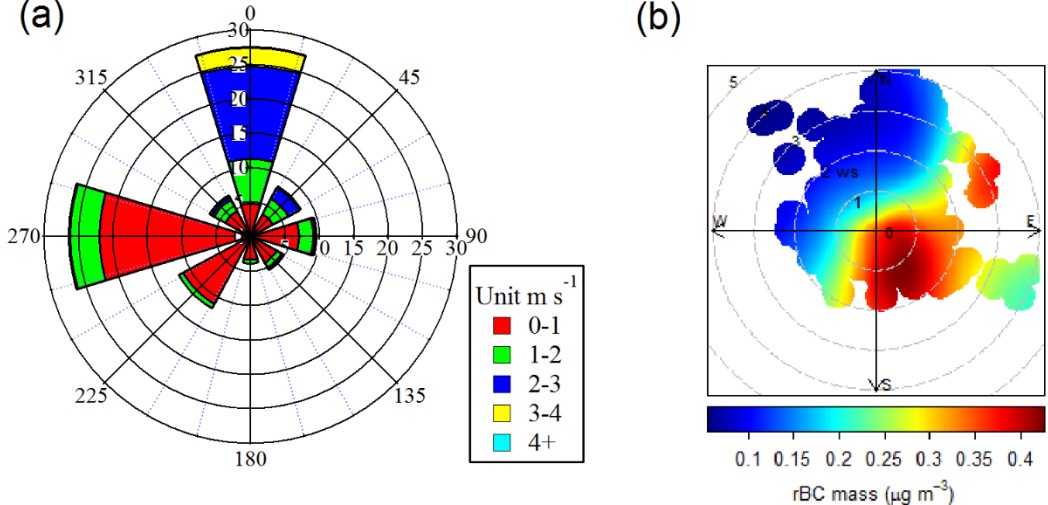

**Figure 4.** (a) Wind rose plot and (b) bivariate polar plot for the rBC mass concentrations based on hourly data.



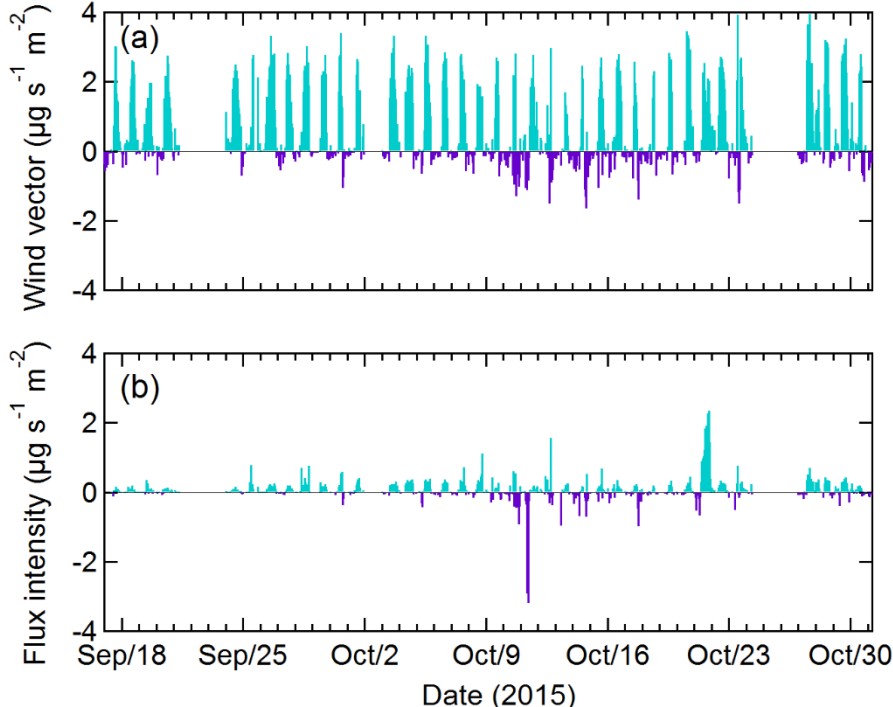

**Figure 5.** Time series of (a) wind vector $(= \frac{1}{n}\sum_{j=1}^{n} WS_j \times cos\theta_j)$ and (b) surface transport intensity for rBC based on the hourly-averaged data at the Lulang site. The positive values indicate the transport direction of rBC from south to north (i.e., from the IGP and Bangladesh to Lulang) and the negative values represent the transport direction of rBC from north to south (i.e., from interior of the Tibetan Plateau to Luang).



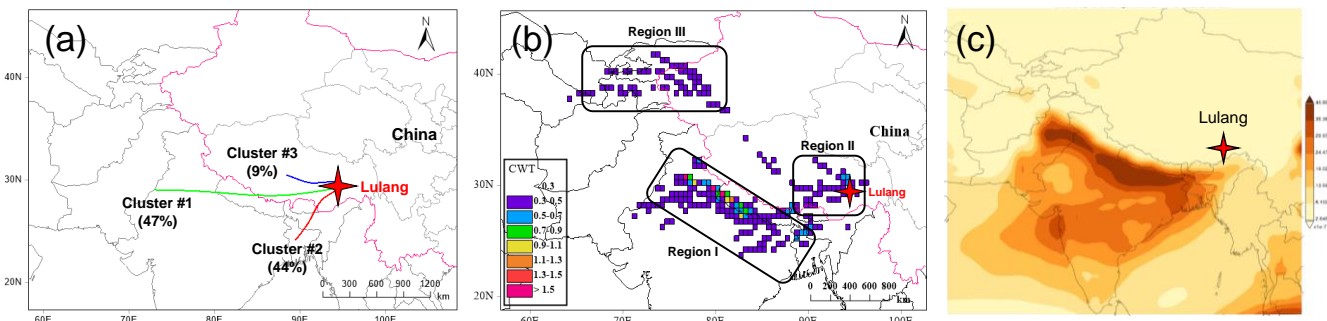

**Figure 6.** The maps of (a) mean trajectory clusters, (b) the concentration-weighted trajectories ($\mu g\ m^{-3}$) for rBC mass concentrations, and (c) the reconstructed BC column mass density ($kg\ m^{-2}$) during the campaign.



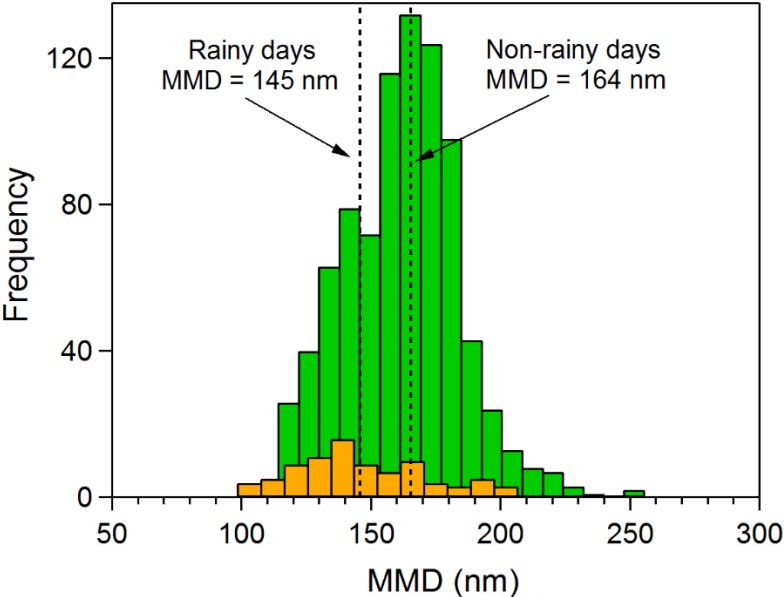

**Figure 7.** Frequency distributions of mass median diameters (MMD) for rainy and non-rainy sampling days. The vertical dash lines denote the average MMDs for those two types of days.



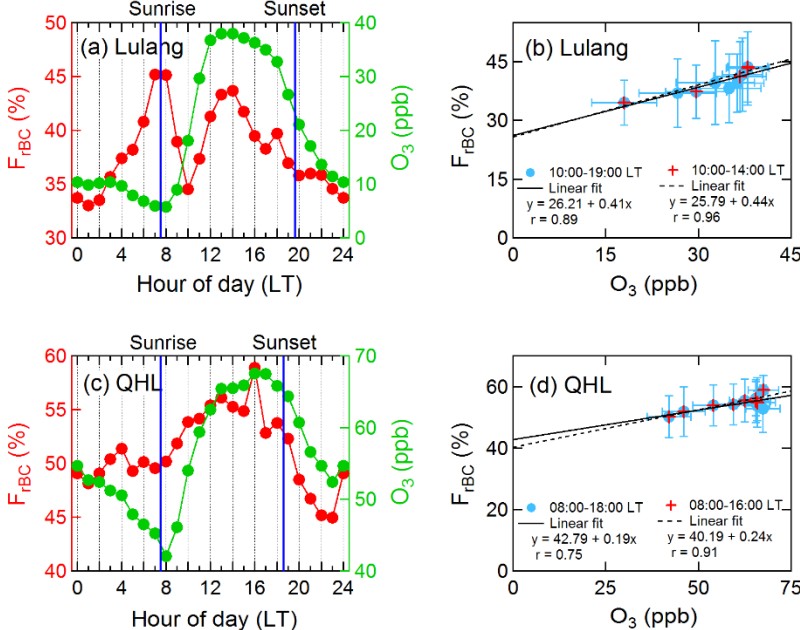

**Figure 8.** (left panels) Diurnal variations of the hourly-averaged number fraction of thickly-coated rBC ($F_{rBC}$) and $O_3$ mixing ratios at Luang and Qinghai Lake (QHL) and (right panels) linear regressions between $F_{rBC}$ and $O_3$ at these two sites.





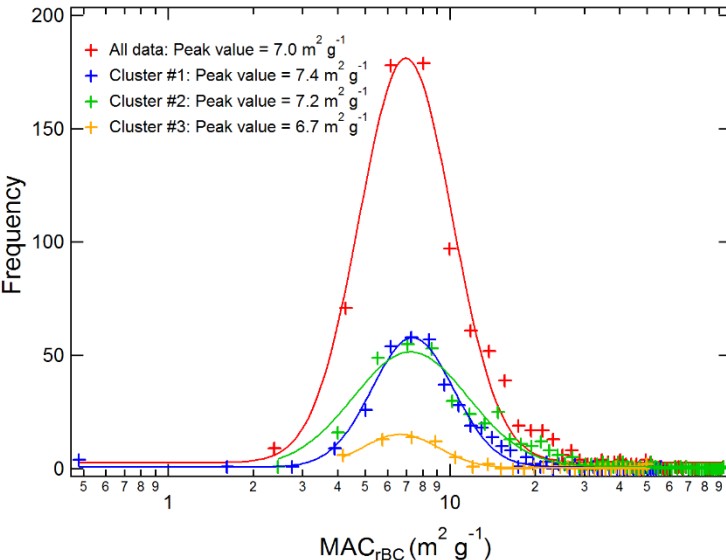

**Figure 9.** Frequency distributions of the mass absorption cross section sof rBC (MAC$_{rBC}$) for the entire campaign and for air masses defined by trajectory cluster.





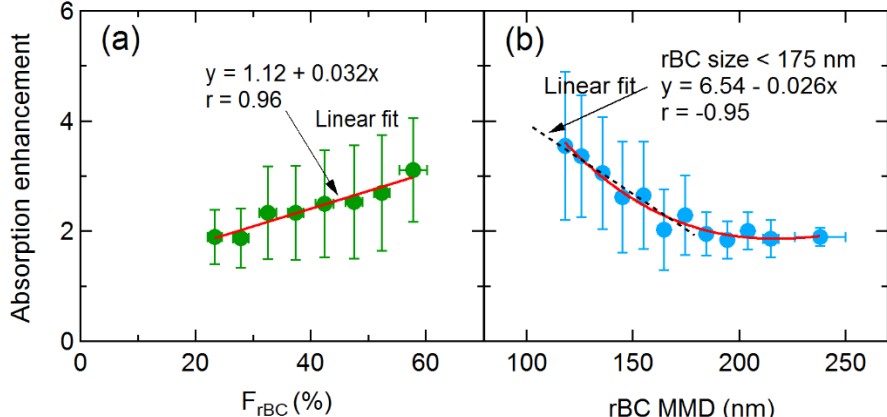

**Figure 10.** Absorption enhancement ($E_{abs}$) versus (a) the number fraction of thickly-coated rBC ($F_{rBC}$) and (b) the mass median diameter (MMD) of rBC during the entire campaign. The error bars correspond to the standard deviations of $E_{abs}$, $F_{rBC}$, and MMD.