# Peer review of "Sources and physicochemical characteristics of black carbon aerosol in the southeastern Tibetan Plateau: internal mixing enhances light absorption"

_Atmospheric Chemistry and Physics, 2017_

## Referee Comment (RC1) · Anonymous Referee #2 · 17 Dec 2017

Totally this MS has been greatly improved. However, there are still some problems in the MS. Firstly, the English still need to be further improved. Some sentences are a little long and need to be cut into short sentences. Secondly, some inconsistent tenses exist in some sentences, please check carefully. This study only cover less than two months in autumn of 2015. Due to radiative and atmospheric chemistry at study area should be different at other seasons, the conclusion of this study should be limited and the uncertainties need also be pointed out in the MS.

---

## Referee Comment (RC2) · Anonymous Referee #4 · 17 Dec 2017

The author has made a sincere effort to improvise the manuscript relative to the previous version. However, there is still a scope to correct it grammatically, the tenses and apt usage of phrases. Below are the primary comments both technical and non-technical which the author can address to enhance the quality of the manuscript.

Page 3 Line 13: It is not clear what high resolution measurements the author is talking, is it about time, space?

Page 6 Line 18-19: Section 2.3.1 requires more justification how evaluation of surface

flux intensity is an indication of regional transport

Page 8 Line 31: Please elaborate few difficulties in obtaining scaling factors

Page 9 Line 1: i) "Fig 3 (a-c) shows the diurnal variations of rBC mass…." Please specify it as the averaged values over the entire campaign. ii) "PBL height" or "PBL depth", check and change throughout the manuscript iii) ..wind speed during the (not over the) entire campaign – check grammatical errors

Line 17: "As shown in Fig. 3 (a–b), the rapid morning increase in rBC was accompanied deepening of the PBL, and therefore, regional transport maybe an important influence on the aerosol populations." The meaning of this sentence is not clear. Reframe the sentence, and what is aerosol populations?

Line 21: "After sunrise, as the PBL starts to deepen, strengthening thermals lift and eventually break the nighttime inversion, and this can lead to the transport of pollutants to the southeastern TP." Reframe the sentence, as PBL expansion is one of the important factor, but not the only factor

Line 30: In this sentence, "The decreasing trend in the late morning…" this change in rBC can well explained using wind direction. Was there any change in wind direction?

Page 12 Line 13: "Fig. S1 shows that rBC core size distribution was well represented by a mono-modal lognormal fit, which is consistent with previous SP2-based observations made across the globe, including urban, rural, remote areas". Consistent with respect to what? Can you further elaborate?

Page 13 Line 1-3: "In contrast, smaller rBC MMDs were found when the polluted air masses came from North India (Cluster #1, 173 nm) or central TP (Cluster #3, 177 nm). Moreover, more aged particles in the plumes tend to be larger than fresher particles from close to the source (Moteki et al., 2007)". So what is the significance of this? Yes aged particle will grow and will be typically larger than fresh ones... What is the std deviation of these mean value? The stated differences are hard to justify (meaning not

significant) without the variation in the mean value?

---

## Referee Comment (RC3) · Anonymous Referee #1 · 4 Jan 2018

\* General comments

The authors have measured rBC mass concentrations, size distributions and mixing states at a remote site on the south-east Tibetan Plateau with a single particle soot photometer. The results are interesting but there are a number of potential issues with the measurements that don't seem to have been addressed, and a number of conclusions are reached without sufficient supporting evidence. Therefore, I have many comments that I believe must be addressed before the manuscript is considered for publication in ACP.

[Figure]

Consiceness is generally a problem. For example, the point that pollution can be transported to the Tibetan Plateau from the south through valleys is made multiple times in different locations with different types of supporting evidence. The consiceness of the manuscript could be improved considerably by making this point only once, with all of the supporting evidence raised at the same time. For example, Figs. 6c and the MODIS images in Fig. S5 make essentially the same point. Why are they discussed at different points in the manuscript?

The large rBC concentrations measured on 21 October is an interesting observation and perhaps worthy of further investigation. Is it possible to link this to a specific event, to see if such events might occur frequently in this region? This is potentially important since the concentrations observed on this day seem to be ~4 times higher than the normal daily peak concentrations.

There are still a number of grammatical errors in the manuscript and sentences that should be split into smaller parts. I have tried to point these out in my specific comments below but I cannot guarantee that this is an exhaustive list. Further proof-reading is required.

* Specific comments

P2, L2: Split the sentence in 2. E.g. '...high-elevation region. It holds ..'

P2, L23: Split the sentence. E.g. '...BC sources are strong (). The TP has become impacted...'

P2, L26: Split the sentence. E.g. '...of surrounding areas. Annually, on average, South and East...'

P3, L4: Artifact should be changed to artefact. And this sentence appears to be conflating two separate concepts. The bulk collection of particles on filters is a design choice, not a measurement artefact. E.g. particles could be classified by size before they are collected on filters. Measurement artefacts are errors arising from the measurement

technique itself, e.g. the filter loading effect.

P3, L11: 'in' missing between 'change' and 'absorption'.

P3, L19: This is not consistent with general practice in the field. E.g. BC mass calculated from measured light absorption is called equivalent BC (eBC), not simply BC (Petzold et al., 2013).

P3, L25: This sentence requires re-wording. What sort of samples? The sentence should also be split. The free-troposphere part deserves its own sentence. Under what conditions are free troposheric air masses sampled at the site?

P4, L6: It should also be mentioned that the SP2 detects elastically scattered light as well as thermal radiation, since the 'scattering signal' is discussed later in the discussion of how SP2 provides mixing state information.

P4, L21: It is not clear how the 20% uncertainty estimate has been arrived at. Please provide further quantitative details. Was the 'SP2 response to ambient rBC mass' determined from an independent measurement of BC mass?

P5, L4: Fig. S1 does not show that the 'vast majority' of rBC particles had VED between 70 and 300 nm. The figure shows that at least approximately half of the rBC particles had VED less than 70 nm (possibly more since the peak of the dN/dlogDp curve has not been reached by 70 nm).

P5, L8: Particle losses in such a Nafion tube (diameter 0.11 inches, length 48 inches) can be very large. Tubes of such small diameter are not typically used in aerosol sampling lines. The authors should demonstrate to what extent particle losses may have affected the measured b_abs values.

P5, L15: b_ext values calculated in this manner require very large aerosol loadings due to the short 0.354 m optical path length in the PAX. The aerosol loadings used for the calibrations should be mentioned along with the range of ambient values measured, to give a sense of where the calibrations have been extrapolated to.

P5, L24: Please provide details of how the ∼15% measurement uncertainty has been arrived at for the PAX. Additionally, the PAX can have difficulty measruring low, ambient absorption levels accurately (e.g. < ∼1 Mm-1). Was a lower limit of detection/quantification used in this study?

P6, L1: Please provide more details on the PBL depths that were used. Are they from a model? It is not clear which data were used by clicking through to the link provided.

P6, L19: It would be useful to discuss what this parameter means physically. I.e. positive values of f indicate ...

P6, L25: Please provide further details about the actual clustering procedure that was used. Currently, only the calculation of the mean angle parameter d_12 is discussed. No information is provided about how this parameter was used to cluster back trajectories into different groups.

P7, L10: This sentence requires more precision. E.g. change to '...the rBC sources that potentially influenced the air sampled at Lulang'

P7, L26: This is difficult to see in Fig. S3. I suggest plotting the frequency distribution on a log x-scale or maybe a reduced x-axis length to highlight this point.

P9, L6: Were data from the 21 October included in the calculation of these diurnal profiles? I'm thinking if the apparent night time peak (or any other feature in the profile) was simply a result of this one-off event of high concentrations.

P9, L18: Missing 'by' between 'accompanied' and 'deepening'.

P9, L27: Change 'south' to 'southern'.

P10, L14: While this is physically plausible, a statistical test should be conducted to determine whether it is really possible to say from this dataset that rBC concentrations were lower on moderate or heavy rain days than rBC concentrations on light rain days. Given the small sample size (only 4 moderate or heavy rain days), it may not be.

P10, L15: Split the sentence. E.g. '... (Fast et al., 2007). Fig. 4a shows...'

P11, L30-35: While interesting, this is the 3rd time this observation has been mentioned. The consiceness of the manuscript could be improved considerably by making this point only once.

P12, L14: An 'and' is required between 'rural' and 'remote'.

P12, L31: Please provide more explanation as to how air mass transport histories might affect rBC size distributions. I'm not aware of any mechanism by which rBC core VED (i.e. rBC mass) would change during transport (e.g. due to evaporation or condensation). Size-dependent removal processes could change the rBC core size distribution as discussed in the sentences following. In any case, the cluster analysis example that is given does not demonstrate that air mass transport histories affected the rBC size distributions. The rBC from the different source regions may simply have had different initial size distributions.

P13, L10: Two things could explain the observed differences in Fig. 7: the absence of long-range transport during the rainy days or preferential wet scavenging of larger rBC cores during the rainy days. For the 2nd hypothesis, could the authors provide a reference or further theoretical argument to indicate whether this is feasible?

P13, L31: Photochemical production of coating material is just one explanation for the increased F_rBC during the afternoon. Mixing layer height was also high during the afternoon. Comparison of Fig. 3 and Fig. 8 suggests F_rBC also correlated well with mixing layer height for the periods from 10:00-19:00. Thus, another potential explanation for the high F_rBC values observed in the afternoon is the mixing of more aged BC particles from aloft to the surface.

P13, L32: I don't think it has been demonstrated that in situ photochemisty is completely responsible for the afternoon increase in F_rBC (see above comment). Therefore, I don't think it is justified to report oxidation rates and compare such rates with

those observed at Qinghai Lake.

P14, L20: The PAX can have difficulties to measure b_abs values less than ~1 Mm-1. Since a considerable amount of the measurements fall in this range, could the authors provide a scatterplot of b_abs vs rBC concentrations from the SP2, and if relevant add it to the supplementary infomation? Such a plot might help to determine if a lower limit of quantification should be applied to the PAX measurements. E.g. if it shows the PAX was insensitive to changes in BC concentration below some threshold.

P14, L26: Given the very narrow nafion drier used in front of the PAX, I think it must be checked whether the b_abs measurements are biased low, which would mean these MAC values are also biased low.

P15, L12: Should be '... if the fraction of thickly coated rBC particles increased by one percent...'

P15, L25: More specifically, if coatings are formed by condensation, this is due to the 1/Diameter dependence of the condensation rate.

P16, Section 4: A number of the conclusions made in this section might need to be updated after the specific comments above have been addressed.

References: Petzold, A., Ogren, J. A., Fiebig, M., Laj, P., Li, S.-M., Baltensperger, U., Holzer-Popp, T., Kinne, S., Pappalardo, G., Sugimoto, N., Wehrli, C., Wiedensohler, A. and Zhang, X.-Y.: Recommendations for reporting "black carbon" measurements, Atmos. Chem. Phys., 13(16), 8365–8379, doi:10.5194/acp-13-8365-2013, 2013.

---

## Author Comment (AC1) · 15 Feb 2018

Totally this MS has been greatly improved. However, there are still some problems in the MS. Firstly, the English still need to be further improved. Some sentences are a little long and need to be cut into short sentences. Secondly, some inconsistent tenses exist in some sentences, please check carefully.

> **Response:** We thank the reviewer for taking the time to review our manuscript. We have revised most of the long sentences to the shorter ones, and the tenses also have been checked carefully. The language has been further proofed by a native English speaker.

This study only cover less than two months in autumn of 2015. Due to radiative and atmospheric chemistry at study area should be different at other seasons, the conclusion of this study should be limited and the uncertainties need also be pointed out in the MS.

> **Response:** In the revised manuscript, we added the following in the conclusion section to clarify the limitation and uncertainties caused by the short sampling period: "We should note that the sources, transport, and radiative effects of the rBC as well as atmospheric conditions likely vary in complex ways with season, and therefore the results from our study (in autumn) are not necessarily representative of other times of the year. Indeed, additional studies need to be conducted to determine how the rBC aerosol at our site and others changes with season."

---

## Author Comment (AC2) · 15 Feb 2018

The author has made a sincere effort to improvise the manuscript relative to the previous version. However, there is still a scope to correct it grammatically, the tenses and apt usage of phrases. Below are the primary comments both technical and non- technical which the author can address to enhance the quality of the manuscript.

**Response:** We thank the reviewer for taking the time to review our manuscript.

Page 3 Line 13: It is not clear what high resolution measurements the author is talking, is it about time, space?

**Response:** The reference was to high time resolution measurements. In the revised manuscript, we revised this sentence to "Accurate information on the physicochemical characteristics of BC can improve our understanding of anthropogenic climate impacts on the TP, but there is still lack of high time resolution measurements on the size and mixing state of BC in this region."

Page 6 Line 18-19: Section 2.3.1 requires more justification how evaluation of surface flux intensity is an indication of regional transport.

**Response:** The transport of pollutants was markedly influenced by meteorological parameters, especially wind speed and wind direction. For local emission sources, wind can facilitate the dilution and dispersion of air pollutants. Strong winds obviously favor dispersion whereas weak winds often lead to the accumulation of air pollutants. For regional sources, strong winds can transport pollutants over long distances, and that can lead to high concentrations of pollutants in downwind areas. If regional transport carried large rBC particles due to high winds, then the calculated surface flux intensity would be large. Therefore, in our study, we viewed the surface flux intensity as a measure of the influence of regional transport in South Asia, and more specifically on the Lulang site using ground-based observations. In the revised manuscript, we added the following text to make it clearer: "Generally, strong winds favor the dispersion of air pollutants for local emission sources whereas weak winds lead to accumulation. In contrast, for

regional sources, strong winds can transport pollutants from upwind areas and cause high concentrations of pollutants downwind. Therefore, in this study, we viewed the surface flux intensity as a measure of the influence of regional transport in South Asia, and more specifically on the Lulang site using ground-based observations."

Page 8 Line 31: Please elaborate few difficulties in obtaining scaling factors

**Response:** Actually, the difficulties in obtaining scaling factors were discussed in our original manuscript; these include the uncertainties caused by the inherent differences instruments themselves and a lack of BC method intercomparisons. This may have been misunderstanding of our original expression, and in the revised manuscript, we changed the sentence to "Therefore, limitations such as those mentioned above make it difficult to establish scaling factors to reconcile the various BC measurements on the TP to a common standard, and direct comparisons of BC data obtained by different methods can be tenuous."

Page 9 Line 1: i) "Fig 3 (a-c) shows the diurnal variations of rBC mass. . ..." Please specify it as the averaged values over the entire campaign. ii) "PBL height" or "PBL depth", check and change throughout the manuscript iii) ...wind speed during the (not over the) entire campaign – check grammatical errors.

**Response:** We revised this sentence in the revised manuscript. It now reads: "Fig. 3 (a–c) shows the diurnal variations of the average rBC mass concentrations, PBL heights, and wind speeds during the campaign." We also changed all the "PBL depth" to "PBL height" throughout the manuscript.

Line 17: "As shown in Fig. 3 (a–b), the rapid morning increase in rBC was accompanied deepening of the PBL, and therefore, regional transport maybe an important influence on the aerosol populations." The meaning of this sentence is not clear. Reframe the sentence, and what is aerosol populations?

**Response:** The "aerosol populations" means different chemical components, such as rBC, organic aerosol, sulfate, nitrate, etc. However, since our study focused on

rBC aerosol, in order to make it more clear, we changed the "aerosol populations" to "rBC particles" in the revised manuscript. We reframed this sentence to "As shown in Fig. 3 (a–b), the rapid morning increases in rBC were accompanied by deepening of the PBL, which suggests the possibility that regional transport had an important influence on rBC particles."

Line 21: "After sunrise, as the PBL starts to deepen, strengthening thermals lift and eventually break the nighttime inversion, and this can lead to the transport of pollutants to the southeastern TP." Reframe the sentence, as PBL expansion is one of the important factor, but not the only factor

**Response:** In the revised manuscript, we revised this expression to "After sunrise, as the PBL starts to deepen, strengthening thermals lift and eventually break up the nighttime inversion. These changes in the atmosphere provide conditions that could support the transport of pollutants to the southeastern TP."

Line 30: In this sentence, "The decreasing trend in the late morning. . ." this change in rBC can well explained using wind direction. Was there any change in wind direction?

**Response:** The prevailing wind directions changed from southwest at 07:00–09:00 to northeast at 10:00–12:00. In the revised manuscript, we added the wind direction discussion. It now reads: "The decreasing trend in rBC loadings in the late morning at Lulang is consistent with the continued deepening of the PBL (Fig. 3b) and the strengthening winds from the northeast (see Fig. 2 and Fig. 3c)."

Page 12 Line 13: "Fig. S1 shows that rBC core size distribution was well represented by a mono-modal lognormal fit, which is consistent with previous SP2-based observations made across the globe, including urban, rural, remote areas". Consistent with respect to what? Can you further elaborate?

**Response:** We revised this sentence in the revised manuscript. It now reads: "Fig. S1 shows that rBC core size distribution was well represented by a mono-modal lognormal fit. This is consistent with the size distributions constructed from previous SP2-based observations made across the globe, including urban, rural,

and remote areas (e.g., Schwarz et al., 2008a; Liu et al., 2010; McMeeking et al., 2011; Huang et al., 2012; Wang et al., 2014)."

Page 13 Line 1-3: "In contrast, smaller rBC MMDs were found when the polluted air masses came from North India (Cluster #1, 173 nm) or central TP (Cluster #3, 177 nm). Moreover, more aged particles in the plumes tend to be larger than fresher particles from close to the source (Moteki et al., 2007)". So what is the significance of this? Yes aged particle will grow and will be typically larger than fresh ones... What is the std deviation of these mean value? The stated differences are hard to justify (meaning not significant) without the variation in the mean value?

**Response:** Here we used the results of cluster analysis as a way of demonstrating that air mass transport histories can affect the size distributions of rBC among different rBC studies. This is because the air masses that originate from different regions can have dissimilar initial rBC size distributions due to their main emission sources, etc. In addition, the rBC core sizes may be changed through coagulation during transport. We added the standard deviation of the mean MMD value for each cluster. We also used t-tests to determine whether there were statistically significant differences in the MMDs from different clusters. In the revised manuscript, we reworked this part. It now reads: "Second, transport histories matter because aging of the particles can affect the size distributions of rBC. Take the cluster analysis as an example: the average rBC MMD was the largest (184  $\pm$ 17 nm) when the polluted air masses originated from central Bangladesh (Cluster #2). In contrast, smaller rBC MMDs were found when the polluted air masses came from North India (Cluster #1,  $173 \pm 26$  nm) or the central TP (Cluster #3,  $177 \pm 19$  nm). These air masses originated from different source regions, and they may have had different rBC sizes initially; but the rBC core sizes also may have changed during transport through coagulation. It should be noted that a t-test for the rBC MMDs from different clusters showed that there was a statistically significant difference between Cluster #1 and #2 (p < 0.01), but was not significant between Cluster #2 and #3 (p = 0.09)."

---

## Author Comment (AC3) · 15 Feb 2018

\*     General comments

The authors have measured rBC mass concentrations, size distributions and mixing states at a remote site on the south-east Tibetan Plateau with a single particle soot photometer. The results are interesting but there are a number of potential issues with the measurements that don't seem to have been addressed, and a number of conclusions are reached without sufficient supporting evidence. Therefore, I have many comments that I believe must be addressed before the manuscript is considered for publication in ACP.

> **Response:** We appreciate the reviewer's thoughtful and valuable comments. We have made most of the changes suggested by the reviewer, both to the text and figures.

Consiceness is generally a problem. For example, the point that pollution can be transported to the Tibetan Plateau from the south through valleys is made multiple times in different locations with different types of supporting evidence. The consiceness of the manuscript could be improved considerably by making this point only once, with all of the supporting evidence raised at the same time. For example, Figs. 6c and the MODIS images in Fig. S5 make essentially the same point. Why are they discussed at different points in the manuscript?

> **Response:** The purpose of showing the MODIS images in Fig. S5 was to show that pollutants (including rBC) from the IGP and Bangladesh were likely transported to Lulang on the morning (08:00–10:00 LT) of sampling days. The Terra satellite passed over the TP region at ~10:30 LT, and the timing of the overpass is within the observed morning peak in rBC. Thus, the MODIS images for the morning combined with the wind distributions are an effective way of illustrating regional transport for a specific case. In contrast, the BC column mass density in Fig. 6c was an average daily distribution for the entire campaign. As the CWT analysis was used for all of the trajectories during the entire campaign, the average daily distribution of BC column mass density was a way of showing the

region with high CWT values corresponded to the large BC loadings. In the revised manuscript, we deleted redundant expressions to make the paper more concise.

The large rBC concentrations measured on 21 October is an interesting observation and perhaps worthy of further investigation. Is it possible to link this to a specific event, to see if such events might occur frequently in this region? This is potentially important since the concentrations observed on this day seem to be ~4 times higher than the normal daily peak concentrations.

**Response:** Although the peak concentration of rBC on 21 October was much higher than the other days, its diurnal profile was similar to most of sampling days; that is, an increasing trend in rBC loadings occurred from 08:00–10:00 LT (see Fig. R1 below). As we discussed in the manuscript, the large morning peaks resulted from the combined effects of local activities and regional transport. Over short time-scales, such as the length of our study, the emission sources can be considered relatively stable. Thus, regional transport was thought to play an important role in this high rBC episode. The three-day backward trajectory analysis for 08:00–10:00 LT on 21 October show that the air parcel was over Guwahati, northeastern India, for a considerable time before arriving at Lulang (see Fig. R2 below). The amount of time the air masses spent over this anthropogenic region was likely sufficient for rBC particle loadings to build up, and it was these particles that were subsequently transported to Lulang. In the revised manuscript, we added the following: "It should be noted that even though the average rBC concentration from 08:00–10:00 on 21 October was ~8 times higher than the average value for other sampling days, the diurnal pattern of 21 October was similar to that seen on other days (Fig. S8a). Indeed, the rBC diurnal loading pattern did not appear to different on this high rBC concentration day (Fig. S8 b and c). Over short time scales, such as the length of our study, one can assume that the local emission sources are relatively stable. Based on the three-day backward trajectory analysis, sudden high rBC loadings such as those on the morning on 21 October may be explained by the slow passage of air over Guwahati

in northeastern India (Fig. S9). Large numbers of rBC particles likely accumulated in the air as it slowly passed over this polluted region, and it was those particles that were eventually transported to Lulang."

[Figure]

**Figure R1.** Diurnal variation of rBC mass concentrations on 21 October, 2015

[Figure]

**Figure R2.** Three-day air-mass trajectories calculated backwards in time for 08:00–10:00 (local time) on 21 October 2015.

There are still a number of grammatical errors in the manuscript and sentences that should be split into smaller parts. I have tried to point these out in my specific comments below but I cannot guarantee that this is an exhaustive list. Further proof-reading is required.

**Response:** In some cases, long sentences are the most effective way of showing relationships among complex concepts, but we have tried to split long sentences into shorter ones. The paper also has been further proofed by a native English speaker.

\*     Specific comments

P2, L2: Split the sentence in 2. E.g. '...high-elevation region. It holds ..'

> **Response:** Change made. It now reads: "The Tibetan Plateau (TP) is the world's largest high-elevation region. It holds the largest ice mass on the planet outside the polar regions and is sometimes called the Earth's "Third Pole" (Yao et al., 2008)."

P2, L23: Split the sentence. E.g. '...BC sources are strong (). The TP has become impacted...'

> **Response:** Change made. It now reads: "Geographically, the TP is surrounded by South and East Asia where BC sources are strong (Zhang et al., 2009), and the TP has become impacted by these high-BC source areas due to the general circulation patterns (Cao et al., 2010; Lu et al., 2012; Zhao et al., 2017)."

P2, L26: Split the sentence. E.g. '...of surrounding areas. Annually, on average, South and East...'

> **Response:** Change made. It now reads: "For example, Lu et al. (2012) found that BC loadings in the Himalayas and TP increased by 41% from 1996 to 2010 due to the influences of surrounding areas. Annually, on average, South and East Asia account for 67% and 17% of BC transported to the plateau, respectively."

P3, L4: Artifact should be changed to artefact. And this sentence appears to be conflating two separate concepts. The bulk collection of particles on filters is a design choice, not a measurement artefact. E.g. particles could be classified by size before they are collected on filters. Measurement artefacts are errors arising from the measurement technique itself, e.g. the filter loading effect.

> **Response:** There is some debate regarding the usage of artifact vs. artifact, but a substantial number of online sources indicate that the difference is just between British vs. American English. Even so, we have made this change. Some online or offline filter-based techniques (e.g., aethalometer and multi-angle absorption

photometer) typically obtain BC mass at a specific size based on the inlet cyclone cutoff diameter. Although cascade impactors have been used to collect size-segregated aerosol samples for BC analysis, these instruments can only obtain data for several size ranges. In order to make this clearer, we revised this sentence in the revised manuscript. It now reads: "Although some aerosol-related field studies have been conducted on the TP, the BC measurements were mainly made using online or offline filter-based techniques (e.g., aethalometer, thermal/optical reflectance method, and multi-angle absorption photometer) (e.g., Engling et al., 2010; Marinoni et al., 2010; Wan et al., 2015; Zhu et al., 2016; Li et al., 2017 ). These techniques are based on the bulk particle deposition onto the filters, and they cannot provide high time resolution information on BC size and mixing state."

P3, L11: 'in' missing between 'change' and 'absorption'.

**Response:** Change made. It now reads: "That study showed that BC particles initially changed from a fractal to spherical morphology with little change in absorption followed by growth into compact particles with large $E_{abs}$."

P3, L19: This is not consistent with general practice in the field. E.g. BC mass calculated from measured light absorption is called equivalent BC (eBC), not simply BC (Petzold et al., 2013).

**Response:** In the revised manuscript, we changed this sentence to "Here the term rBC is used exclusively in reference to SP2 measurements while eBC (equivalent BC) and EC (elemental carbon) refer to the data from the optical absorption method and the thermal heating and optical absorption techniques, respectively, used in other studies (Petzold et al., 2013)."

P3, L25: This sentence requires re-wording. What sort of samples? The sentence should also be split. The free-troposphere part deserves its own sentence. Under what conditions are free troposheric air masses sampled at the site?

**Response:** Previous studies based on meteorological analysis have identified the Tibetan Plateau and the Himalayas as global hot spots for deep stratosphere-totroposphere transport (e.g., Škerlak et al., 2014). Moreover, measurements in those areas can at times reflect the composition of free tropospheric air. As it was beyond the scope of our study to identify the conditions that would lead to the sampling tropospheric air, we revised this text to, "Physicochemical and optical properties of rBC aerosol were measured in samples collected from a remote area of Lulang, which is located on the southeastern part of the TP (Fig. 1). An intensive measurement campaign was conducted from 17 September to 31 October 2015 on the dormitory rooftop of the Integrated Observation and Research Station for Alpine Environment in South-East Tibet, Chinese Academy of Sciences (94.44°E, 29.46°N, ~3300 m above sea level)."

**Reference:**

Škerlak, B., Sprenger, M., and Wernli, H.: A global climatology of stratosphere–troposphere exchange using the ERA-Interim data set from 1979 to 2011, Atmos. Chem. Phys., 14, 913–937, doi:10.5194/acp-14-913-2014, 2014.

P4, L6: It should also be mentioned that the SP2 detects elastically scattered light as well as thermal radiation, since the 'scattering signal' is discussed later in the discussion of how SP2 provides mixing state information.

**Response:** Following the reviewer's suggestion, we added the following in the revised manuscript: "Simultaneously, the laser light scattered by the rBC-containing particle was detected elastically."

P4, L21: It is not clear how the 20% uncertainty estimate has been arrived at. Please provide further quantitative details. Was the 'SP2 response to ambient rBC mass' determined from an independent measurement of BC mass?

**Response:** The SP2 needs an empirical calibration to retrieve the rBC mass from the incandescence signal, and the sensitivity of the SP2 differs among BC particle types. Ideally, for atmospheric studies, the SP2 should be calibrated using ambient particles containing a known mass of rBC. However, such "ambient BC" calibration particles cannot easily be obtained. Thus, commercially available BC

particles are commonly used for SP2 calibration instead. In the study of Laborde et al. (2012), the sensitivity of the SP2 to different BC types was tested to characterize the potential error introduced by using non-ambient BC for calibration. We cited their results (~15%) as the uncertainty of the SP2 response to ambient rBC mass in our study. The propogated uncertainty of the SP2 measurement was estimated from the square root of uncertainties caused by the SP2 response to ambient rBC mass (~15%), sample flow (10%), and estimates of the rBC mass beyond of SP2 detection range (10%). In the revised manuscript, we revised the original sentence to "The uncertainty of the SP2 mass measurements was ~20%, which was estimated by propagating the uncertainties caused by the SP2 response to ambient rBC mass (~15%, Laborde et al., 2012), sample flow (10%), and estimates of the rBC mass beyond the SP2 detection range (10%)."

**Reference:**

Laborde, M., Mertes, P., Zieger, P., Dommen, J., Baltensperger, U., and Gysel, M.: Sensitivity of the Single Particle Soot Photometer to different black carbon types, Atmos. Meas. Tech., 5, 1031-1043, doi:10.5194/amt-5-1031-2012, 2012.

P5, L4: Fig. S1 does not show that the 'vast majority' of rBC particles had VED between 70 and 300 nm. The figure shows that at least approximately half of the rBC particles had VED less than 70 nm (possibly more since the peak of the dN/dlogDp curve has not been reached by 70 nm).

**Response:** We meant that the size range of 70–300 nm accounted for most of the detected rBC particles. Due to limitations of the SP2 measurements, we could not obtain data for rBC particles smaller than 70 nm. In the revised manuscript, we modified this sentence to "An examination of the number size distribution of rBC shows that this was not a critical limitation in the following analysis because that size range contained the vast majority of the detected rBC particles (see Fig. S1)."

P5, L8: Particle losses in such a Nafion tube (diameter 0.11 inches, length 48 inches) can be very large. Tubes of such small diameter are not typically used in aerosol

sampling lines. The authors should demonstrate to what extent particle losses may have affected the measured b_abs values.

**Response:** We conducted an experiment to compare the $b_{abs}$ measured with and without the Nafion tube (Perma Pure MD-700 dryer). As shown in Fig. R3 below, the particle loss for this type of Nafion tube may be ~10%. Thus, the $b_{abs}$ values were scaled up by a factor of ~1.1 to compensate for the losses. We have added this information in the revised manuscript.

[Figure]

**Figure R3.** Scatter plot of light absorption coefficient measured with ($b_{abs\_nafion}$) and without ($b_{abs\_without\ nafion}$) Nafion dryer (MD-110-48S).

P5, L15: b_ext values calculated in this manner require very large aerosol loadings due to the short 0.354 m optical path length in the PAX. The aerosol loadings used for the calibrations should be mentioned along with the range of ambient values measured, to give a sense of where the calibrations have been extrapolated to.

**Response:** In the revised manuscript, we added a Fig. S3 (also see Fig. R4 below) to show the results of PAX calibration. Following the reviewer's suggestion, we added the following in the revised manuscript: "Different concentration gradients of freshly-generated propane soot were used to give an absorption reading of ~10 to 16700 $Mm^{-1}$ for absorption calibration (Fig. S3)."

[Figure]

**Figure R4.** Scattering and absorption calibration of the photoacoustic extinctiometer ($PAX_{870}$).

P5, L24: Please provide details of how the ~15% measurement uncertainty has been arrived at for the PAX. Additionally, the PAX can have difficulty measruring low, ambient absorption levels accurately (e.g. $< \sim 1$ $Mm^{-1}$). Was a lower limit of detection/quantification used in this study?

> **Response:** We re-analyzed the PAX data and discarded the data that had values $<$ 1 $Mm^{-1}$ (~15% of total number of $b_{abs}$ measurements), and the results were changed accordingly in the revised manuscript. Moreover, we added the following text to clarify how the uncertainty obtained: "The uncertainty of the PAX for absorption measurements was estimated to be ~15% based on the variations of $b_{abs}$ caused by the noise during the sampling period."

P6, L1: Please provide more details on the PBL depths that were used. Are they from a model? It is not clear which data were used by clicking through to the link provided.

> **Response:** The PBL heights were simulated from the European Centre for Medium-range Weather Forecasts (ECMWF) model. One can download the PBL data directly from ERA-Interim (Jan. 1979–present) reanalysis datasets at http://apps.ecmwf.int/datasets. In the revised manuscript, we changed the original sentence to "The planetary boundary layer (PBL) heights were obtained from the European Centre for Medium-range Weather Forecasts (ECMWF). These can be downloaded from ERA-Interim (Jan. 1979–present) reanalysis datasets at http://apps.ecmwf.int/datasets."

P6, L19: It would be useful to discuss what this parameter means physically. I.e. positive values of f indicate ...

**Response:** Following the reviewer's suggestion, we added the following: "Positive values for f were considered indicative of transport from outside the TP (e.g., the Indo-Gangetic Plain, IGP, and Bangladesh) whereas negative values indicated transport from the interior of the TP."

P6, L25: Please provide further details about the actual clustering procedure that was used. Currently, only the calculation of the mean angle parameter d_12 is discussed. No information is provided about how this parameter was used to cluster back trajectories into different groups.

**Response:** Following the reviewer's suggestion, we added the following in the revised manuscript: "A two-step algorithm was used to produce the clusters. First, a Hartigan's K mean algorithm was used to construct several clusters of backward trajectories. Those clusters were then examined visually, and selected backward trajectories were moved from one cluster to another in order to define clusters that were easier to interpret with respect to geographical and/or anthropogenic source regions. In this study, three clusters were chosen as representative of the backward trajectory clusters. The simulation was conducted using the GIS-based TrajStat software (Wang et al., 2009)."

P7, L10: This sentence requires more precision. E.g. change to '...the rBC sources that potentially influenced the air sampled at Lulang'

**Response:** Following the reviewer's suggestion, we revised the sentence to: "A CWT model was used to construct the spatial distribution of the rBC sources that potentially influenced the air sampled at Lulang."

P7, L26: This is difficult to see in Fig. S3. I suggest plotting the frequency distribution on a log x-scale or maybe a reduced x-axis length to highlight this point.

**Response:** We have replotted Fig. S3 in the revised supporting information. The new Fig. S5 is shown below (Fig. R5):

[Figure]

**Figure R5.** Frequency distribution of rBC mass concentrations during the campaign.

P9, L6: Were data from the 21 October included in the calculation of these diurnal profiles? I'm thinking if the apparent night time peak (or any other feature in the profile) was simply a result of this one-off event of high concentrations.

**Response:** In our original manuscript, the diurnal variations of rBC included the data from 21 October. As shown in Fig. R6 below, the diurnal pattern was similar with and without the data from 21 October. Following the reviewer's suggestion above, we have added some discussion about this high rBC episode in our revised manuscript. Please see the response above.

[Figure]

**Figure R6.** Diurnal variations of rBC with and without data from October 21.

P9, L18: Missing 'by' between 'accompanied' and 'deepening'.

**Response:** Change made. It now reads: "As shown in Fig. 3 (a–b), the rapid morning increases in rBC were accompanied by deepening of the PBL, which suggests the possibility that regional transport had an important influence on rBC particles."

P9, L27: Change 'south' to 'southern'.

**Response:** Change made. It now reads: "The true color images reveal obvious pollution bands along the IGP and Bangladesh that piled up on the southern margin of the TP."

P10, L14: While this is physically plausible, a statistical test should be conducted to determine whether it is really possible to say from this dataset that rBC concentrations were lower on moderate or heavy rain days than rBC concentrations on light rain days. Given the small sample size (only 4 moderate or heavy rain days), it may not be.

**Response:** A t-test for the rBC concentrations during light and strong rains showed that there was a statistically significant difference between them at a probability for chance occurrence of $p < 0.01$ ($p = 4.5 \times 10^{-6}$). We added a sentence to this effect in the revised manuscript. It reads "A t-test for the rBC concentrations during light and strong rains showed that there was a statistically significant difference between them at a probability for chance occurrence of $p < 0.01\%$."

P10, L15: Split the sentence. E.g. '... (Fast et al., 2007). Fig. 4a shows...'

**Response:** Change made. It now reads: "Wind speed and wind direction play crucial roles in the dilution and dispersion of pollutants (Fast et al., 2007). Fig. 4a shows the wind speeds and directions during the study."

P11, L30-35: While interesting, this is the 3rd time this observation has been mentioned. The consiceness of the manuscript could be improved considerably by making this point only once.

**Response:** In the revised manuscript, we deleted the redundant expressions to make the discussion more concise. Please see the new manuscript.

P12, L14: An 'and' is required between 'rural' and 'remote'.

**Response:** Change made. It now reads: "Fig. S1 shows that rBC core size distribution was well represented by a mono-modal lognormal fit. This is consistent with the size distributions constructed from previous SP2-based observations made across the globe, including urban, rural, and remote areas (e.g., Schwarz et al., 2008a; Liu et al., 2010; McMeeking et al., 2011; Huang et al., 2012; Wang et al., 2014)."

P12, L31: Please provide more explanation as to how air mass transport histories might affect rBC size distributions. I'm not aware of any mechanism by which rBC core VED (i.e. rBC mass) would change during transport (e.g. due to evaporation or condensation). Size-dependent removal processes could change the rBC core size distribution as discussed in the sentences following. In any case, the cluster analysis example that is given does not demonstrate that air mass transport histories affected the rBC size distributions. The rBC from the different source regions may simply have had different initial size distributions.

**Response:** The effects of air mass transport histories on the aerosol populations are mainly due to two things: one is where they originate, and the other is the effects of atmospheric processing during transport. We agree with the reviewer that different source regions may affect the initial size distributions of rBC. With reference to atmospheric processing, the growth of particles in ambient air can be affected by water accretion, coagulation, vapor condensation, and addition of materials formed through heterogeneous reactions. All of these can lead to rBC-containing particle growth as measured by aerodynamic size. However, only the process of coagulation can make the "rBC core" in a particle grow, i.e., increase the rBC in VED. In the revised manuscript, we changed the original expression to make this clear. It now reads: "Second, transport histories matter because aging of

the particles can affect the size distributions of rBC. Take the cluster analysis as an example: the average rBC MMD was the largest (184 ± 17 nm) when the polluted air masses originated from central Bangladesh (Cluster #2). In contrast, smaller rBC MMDs were found when the polluted air masses came from North India (Cluster #1, 173 ± 26 nm) or the central TP (Cluster #3, 177 ± 19 nm). These air masses originated from different source regions, and they may have had different rBC sizes initially; but the rBC core sizes also may have changed during transport through coagulation. It should be noted that a t-test for the rBC MMDs from different clusters showed that there was a statistically significant difference between Cluster #1 and #2 ($p < 0.01$), but was not significant between Cluster #2 and #3 ($p = 0.09$)."

P13, L10: Two things could explain the observed differences in Fig. 7: the absence of long-range transport during the rainy days or preferential wet scavenging of larger rBC cores during the rainy days. For the 2nd hypothesis, could the authors provide a reference or further theoretical argument to indicate whether this is feasible?

**Response:** Following the reviewer's suggestion, we revised the original expression to "Compared with non-rainy days, the smaller rBC on rainy days can be explained by the absence of long-range transport and by the preferential wet scavenging of larger rBC cores (Taylor et al., 2014)."

P13, L31: Photochemical production of coating material is just one explanation for the increased F_rBC during the afternoon. Mixing layer height was also high during the afternoon. Comparison of Fig. 3 and Fig. 8 suggests F_rBC also correlated well with mixing layer height for the periods from 10:00-19:00. Thus, another potential explanation for the high F_rBC values observed in the afternoon is the mixing of more aged BC particles from aloft to the surface.

**Response:** We added this possible explanation in the revised manuscript: "Moreover, the variations in $F_{rBC}$ during the daytime at Lulang also covaried with

the PBL heights, indicating that aged rBC particles may have been transported from aloft to the surface."

P13, L32: I don't think it has been demonstrated that in situ photochemisty is completely responsible for the afternoon increase in F_rBC (see above comment). Therefore, I don't think it is justified to report oxidation rates and compare such rates with those observed at Qinghai Lake.

**Response:** As we were not able to quantify the effects of aged rBC particles throughout the column, we deleted the discussion of the oxidation rates in the revised manuscript.

P14, L20: The PAX can have difficulties to measure b_abs values less than ~1 Mm$^{-1}$. Since a considerable amount of the measurements fall in this range, could the authors provide a scatterplot of b_abs vs rBC concentrations from the SP2, and if relevant add it to the supplementary information? Such a plot might help to determine if a lower limit of quantification should be applied to the PAX measurements. E.g. if it shows the PAX was insensitive to changes in BC concentration below some threshold.

**Response:** We reanalyzed the PAX data. There was no correlation between $b_{abs}$ and rBC mass concentrations when $b_{abs}$ values less than 1 Mm$^{-1}$. During our sampling period, ~15% of total number of $b_{abs}$ observations was lower than the minimum detection limit of 1.0 Mm$^{-1}$. These data were excluded in the revised manuscript. We have added this information in our revised Section 2.2.2, and the results in Section 3.5 (both text and figures) also has been reworked accordingly.

P14, L26: Given the very narrow nafion drier used in front of the PAX, I think it must be checked whether the b_abs measurements are biased low, which would mean these MAC values are also biased low.

**Response:** As shown in Fig. R3 above, this type of nafion dryer may cause ~10% of the loss for light-absorbing particles. In the revised manuscript, the values of $b_{abs}$ were scaled up by a factor of ~1.1 to compensate for the losses, and the results in Section 3.5 (both text and figures) has been reworked accordingly.

P15, L12: Should be '... if the fraction of thickly coated rBC particles increased by one percent...'

**Response:** Change made. It now reads: "This means that if the fraction of thickly-coated rBC particles increased by one percent, the rBC particles would absorb 3% more light."

P15, L25: More specifically, if coatings are formed by condensation, this is due to the 1/Diameter dependence of the condensation rate.

**Response:** We added this explanation in the revised manuscript. It now reads: "The variations in $E_{abs}$ were relatively constant for rBC MMD > 170 nm. When coatings form by condensation, a 1/diameter dependence would apply to the condensation rate. Thus, larger rBC cores have smaller degree of internal mixing and weaker absorption amplification than smaller cores on the one hand, but on the other hand, larger rBC core size also would decrease the $MAC_{rBC,uncoated}$ according to the Mie model (see the relationship between $MAC_{rBC,uncoated}$ and MMD in Fig. S12)."

P16, Section 4: A number of the conclusions made in this section might need to be updated after the specific comments above have been addressed.

**Response:** We revised the conclusions accordingly. Please see the conclusion section in the revised manuscript.

References: Petzold, A., Ogren, J. A., Fiebig, M., Laj, P., Li, S.-M., Baltensperger, U., Holzer-Popp, T., Kinne, S., Pappalardo, G., Sugimoto, N., Wehrli, C., Wiedensohler, A. and Zhang, X.-Y.: Recommendations for reporting "black carbon" measurements, Atmos. Chem. Phys., 13(16), 8365–8379, doi:10.5194/acp-13-8365-2013, 2013.